# Metagenome-guided culturomics for the targeted enrichment of gut microbes

Jeremy Armetta[1,3], Simone S. Li [1,2,4,5], Troels Holger Vaaben[1], Ruben Vazquez-Uribe [1,6] & Morten O. A. Sommer [1] ✉

The gut microbiome significantly impacts human health, yet cultivation challenges hinder its exploration. Here, we combine deep whole-metagenome sequencing with culturomics to selectively enrich for taxa and functional capabilities of interest. Using a modified commercial base medium, 50 growth modifications were evaluated, spanning antibiotics, physico-chemical conditions, and bioactive compounds. Whole-metagenome sequencing identified medium additives, like caffeine, that enhance taxa often associated with healthier subjects (e.g., *Lachnospiraceae, Oscillospiraceae, Ruminococcaceae*). We also explore the impact of modifications on the composition of cultured communities and establish a link between medium preference and microbial phylogeny. Leveraging these insights, we demonstrate that combinations of media modifications can further enhance the targeted enrichment of taxa and metabolic functions, such as *Collinsella aerofaciens*, or strains harboring biochemical pathways involved in dopamine metabolism. This streamlined, scalable approach unlocks the potential for selective enrichment, advancing microbiome research by understanding the impact of different cultivation parameters on gut microbes.

The gut microbiome has emerged as an important factor in human physiology, influencing health across vital processes, including development, metabolism, immunity, and neurological signaling[1–4]. The extensive genetic repertoire of the gut microbiota surpasses the human genome by 150-fold[5], and its potential metabolic capacity rivals that of the liver[6], with a potential to regulate up to 10% of the human transcriptome[7]. A healthy gut microbiome is characterized by species of diverse taxonomy that harbor a rich gene pool, and functional redundancy[8]. While culture-independent studies have revealed the diversity of gut microbiome, cultivation remains essential to enable study of microbial mechanisms, validate ecological interactions, and the development of Next-Generation Probiotics[9,10] and Advanced Microbiome Therapeutics (AMT)[11,12].

Recent estimates suggest that the gastrointestinal tract of an individual may include more than 400 different microbial species[13].

However, only a fraction of these have been successfully cultured under controlled conditions, underscoring the need for advancements in cultivation-based approaches[14]. Challenges to cultivating gut microbiota in the laboratory stem from their complex growth requirements, including sensitivity to oxygen[15,16], and compounded by co-cultivation needs with other gut microbiota species[16].

To address this, culturomics integrates large-scale -omics approaches, particularly metagenomics, with high-throughput cultivation methods[17]. While most culturomics studies have focused on the cultivation of novel microorganisms[18–21], the quantity of culture conditions may not necessarily align with the amount of unique species recovered. Indeed, a positive correlation exists between a microbe's culturability and its relative abundance in the original sample[22]. Isolation of important keystone taxa or slow-growing species, often found in low abundance within a sample, prompted the development of

[1]Novo Nordisk Foundation Center for Biosustainability, Technical University of Denmark Kgs., Lyngby, Denmark. [2]School of Chemistry and Molecular Biosciences, The University of Queensland, St Lucia, QLD, Australia. [3]Present address: Novonesis A/S, Hørsholm, Denmark. [4]Present address: Department of Microbiology, Biomedicine Discovery Institute, Monash University, Melbourne, VIC, Australia. [5]Present address: Centre to Impact Antimicrobial Resistance, Monash University, Melbourne, VIC, Australia. [6]Present address: Center for Microbiology, VIB, Leuven, Belgium. ✉e-mail: msom@bio.dtu.dk

advanced technologies like complex microfluidics[23] and cell sorting approaches with molecular tags[24,25]. However, these solutions often require substantial resources and sophisticated setups that may not be accessible to all researchers. Scalable high-throughput approaches are, therefore, crucial to advance cultivation-driven microbiome research.

The use of selective media in targeted cultivation and enrichment approaches offers an opportunity to enhance the growth of desirable microbes while restricting other fast-growing species[26]. This strategy enables preferential cultivation of microbes of interest, which is especially useful to detect low-abundance or slow-growing species, such as pathogens relevant to medical microbiology[27]. Implementing this technique requires an understanding of the metabolic needs of specific taxa in order to guide efforts to enhance the growth of target taxa and/or hinder more abundant, fast-growing taxa through regulating parameters of the culture media, including temperature, pH, specific carbon sources, and inhibitory compounds[27–31].

Antibiotics are frequently used as inhibitors in cultivation experiments, particularly targeting fastidious microbes[32]. Rettedal et al. successfully used antibiotics and phenotypic profiling to isolate uncultivated bacteria from the gut microbiota[32,33]. However, these methods may inadvertently favor the selection of multi-drug resistant strains[34]. Furthermore, most culture media were designed in the previous century, and do not facilitate growth of low-abundance or slow-growing gut microbes. To this end, selective media that replicate conditions of the human gastrointestinal tract, such as pH and food-derived carbon sources, remain an underexplored research area.

Here, we present an approach that combines deep whole meta-genome sequencing with culturomics to identify growth conditions that allow for targeted enrichment and enhanced cultivation of selected gut microbes and strains with desired functional capabilities. Designed with the broader research community in mind, we evaluate numerous growth conditions using a commercially available medium in Petri dishes, thus providing an avenue for the rapid and selective enrichment of microorganisms of interest, and support further advancements in microbiome research.

## Results

### Design principles for targeted enrichment in gut microbial communities

To fully unlock the potential of cultivation-driven microbiome research, the cultivation methods must be broadly applicable. To achieve this goal, our approach employs a commercially available medium as a base (Gifu Anaerobic Medium, GAM)[32,35–37]. We modified the medium's composition by introducing hemin, vitamin K1, and antioxidants to enhance the recovery of fastidious gut microbes[38–40]. This modified recipe was used as the basis for all experiments and is referred to as "Base medium" throughout the paper.

To explore the impact of different growth parameters on the cultivation of gut microbes, 50 modifications of growth parameters from six categories were selected to modulate the composition and occurrence of communities growing on the plates (Fig. 1). These modifications were chosen for their documented effects on bacterial

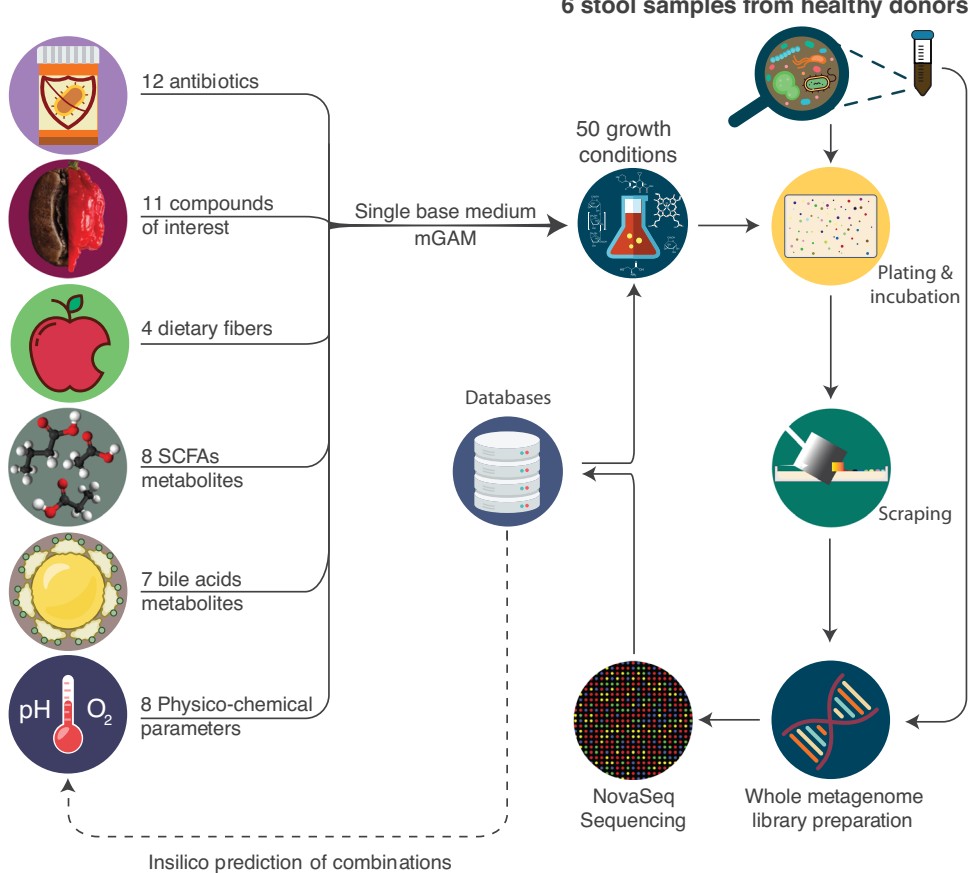

**6 stool samples from healthy donors**

- 12 antibiotics
- 11 compounds of interest
- 4 dietary fibers
- 8 SCFAs metabolites
- 7 bile acids metabolites
- 8 Physico-chemical parameters

Single base medium mGAM → 50 growth conditions → Plating & incubation → Scraping → Whole metagenome library preparation → NovaSeq Sequencing → Databases

Insilico prediction of combinations for targeted enrichment

**Fig. 1 | Overview of the experimental design.** 50 growth medium modifications were selected for our initial single modification cultivations. Each additive or modification was applied to the base medium: a modified Gifu Anaerobic Medium (mGAM). Stool samples from six healthy donors were plated on each medium modification using two dilutions and 612 (6 donors × 51 conditions × 2 dilutions) plates were incubated anaerobically. After seven days, one dilution plate out of the two created for each sample was scraped to recover the biomass and prepared for sequencing on an Illumina NovaSeq.

metabolism or potential interactions with gut microbes due to gastrointestinal tract physiology or host diet.

The first group is composed of 12 antibiotics from different classes selected for their different modes of action, known targets, broad spectrum, and potency[32,41]. The second modification group contains compounds of interest that are commonly present in the gastrointestinal tract (urea, ethanol, sodium chloride, oxalate), aromatic amino acids involved in secondary metabolism of gut microbes (tryptophan, histidine, tyrosine), alkaloids frequently present in hosts' diets (capsaicin, caffeine), and compounds known to have a bactericidal effect (fluoride[42], gold nanoparticles[43]). The third group consists of complex carbohydrate substrates, including three dietary fibers and one highly glycosylated protein, mucin, which is found throughout the gastrointestinal tract. While mucin is not a carbohydrate, it can be used to selectively favor the growth of specialized gut microbes similarly to other complex polysaccharides, such as pectin, inulin and xanthan gum[44], as they can only be metabolized by specialized microorganisms[45]. The fourth group includes eight short-chain fatty acids (SCFAs); these molecules are the metabolic product of gut microbes in the gastrointestinal tract that can act as potent inhibitors by changing the pH[46] or benefit the central metabolism of some bacterial species[35]. The fifth group contains seven primary and secondary bile acids, host-derived metabolites known to modulate the gut microbiota and host physiology[47]. While knowledge about the impact of bile acids on communities in cultivation studies is still limited[36], conjugated primary bile acids such as taurocholic acid are known to increase the culturability of spore-forming bacteria by up to 70,000-fold[31]. Finally, the sixth group is composed of physicochemical modifications with variations in incubation temperature, pH, media dilution, or the presence of oxygen. Lower incubation temperatures, such as 30 °C, can improve the recovery of slow-growing bacteria through longer generation times[36], while dilution of the base medium could enable the recovery of a higher diversity of unique species[32].

## Media modifications enable selective cultivation

Microbial communities from stool samples of six healthy human donors were cultured in Petri dishes containing the base medium and each of the 50 modifications. After incubation, colonies from the plates were scraped, and shotgun metagenomes were obtained for cultured communities, as well as the original stool sample. Variations in biomass and morphological diversity were observed across modifications and donors, with some plates showing low biomass (chloramphenicol, DCA) and others robust growth (pectin, inulin). This suggests modification-dependent effects on microbial community growth, given the shared use of the base medium. This was also reflected in DNA extraction yields and sequencing read counts (Supplementary Fig. 1).

The study identified a diverse taxonomic composition of 334 known and putative species (Metagenomic operational taxonomic units (mOTUs)) across ten phyla (Fig. 2A, B). Firmicutes dominated, followed by Bacteroidetes, Actinobacteria, and Proteobacteria, in line with previous observations[48,49]. Most diversity was exclusive to the original stool samples, with 149 mOTUs absent in single modification cultivations. Overall, our single modification cultivations recovered 42% of species that were detected in the stool samples (105 shared mOTUs). Importantly, 80 mOTUs were solely discovered in culture, including Fusobacteria and many Firmicutes species. These findings, consistent with culturomics studies[17,28], indicated that our cultivation approach can support growth of diverse taxa that were undetected at the depth of sequencing used in our cultivation-independent surveys. Notably, the base medium captured 51% (54/105 mOTUs) of the taxonomic diversity shared between stool and cultivation, with 24% (19/80) of mOTUs unique to cultivation, supporting the notion that a single medium modification can drive taxonomic diversity of recovered microbes.

The number of mOTUs detected per sample ranged from 0 to 50, averaging 21.3 OTUs (SD = 8.3) per modification in our single modification cultivations (Fig. 3A). There were differences in richness based on donors, with donor Pr3 consistently having higher OTU richness across modifications, and Pr5 and Pr6 showing lower richness. However, these differences did not correlate with the richness observed in the original stool sample. This suggests that using stool richness as a proxy may not accurately estimate OTU richness in cultivation studies. Despite variations in OTUs between stool and cultivations, the overall number of microbial species we detected in stool was, on average, similar to the total observed in all media modifications.

The original stool samples exhibited higher phylogenetic diversity (PD) that corresponded to increased mOTU richness (Fig. 3B). Overall, while some variation in PD was observed across donors due to differences in inter-donor initial composition, the media modifications had a marked influence in driving PD changes. In particular, conditions such as histidine, chloramphenicol, vancomycin, cholic acid (CA), glycocholic acid (GCA), caffeine, ethanol, fluoride, nanoparticles, 10X dilution, and pH4 were consistently associated with increased PD across different donors. Conversely, treatments like clindamycin, tetracycline, chenodeoxycholic acid (CDCA), deoxycholic acid (DCA), taurocholic acid (TCA), sodium salts, and aerobic incubation resulted in the lowest PD, indicating that these conditions led to a selection of fewer, more phylogenetically similar OTUs.

Relative abundance was dominated by members of the Bacteroidetes and Proteobacteria phyla across most modifications (Supplementary Fig. 1), with exceptions in modifications such as pH5, ethanol, sodium salt, nanoparticles, caffeine, inulin, taurine, vancomycin, clindamycin, ciprofloxacin, chloramphenicol, and cefotaxime, where a reduction in relative abundance of families within these phyla was observed (Fig. 3C). Interestingly, these media modifications also achieved the largest phylogenetic diversity, often from Firmicutes and Verrucomicrobiota, suggesting that these specific modifications can disrupt the dominance of more abundant taxa, such as Bacteroidetes or Proteobacteria, allowing less abundant taxa to proliferate.

In the aromatic amino acids group, we observed that histidine and tryptophan tended to enrich for more diverse taxa than tyrosine, with tryptophan appearing to selectively favor Clostridiaceae, Bifidobacteriaceae, and Akkermansiaceae species compared to the base medium (as suggested by PD values in Fig. 3B and the number of green squares in Fig. 3C). However, these observations are only qualitative and merit further exploration. Antibiotics generally resulted in lower phylogenetic diversity, particularly clindamycin and tetracycline, which detected on average three and four families, respectively (Fig. 3C). Interestingly, imipenem, niclosamide, and piperacillin, despite different modes of action, showed similar enrichment profiles. Furthermore, bile acids exhibited diverse enrichment patterns, taxa belonging to Eggerthellaceae, Clostridiaceae, and Bifidobacteriaceae seemed to be positively influenced in cultures containing CA, GCA, and TCA. SCFAs modifications showed similar profiles at the family level, with some notable observations, such as a possible enrichment for Clostridiaceae in butyrate or Eggerthellaceae in formate.

Lachnospiraceae, Oscillospiraceae, and Ruminococcaceae families, known for being associated with healthier individuals[19], were selectively enriched in multiple conditions, such as tryptophan, cefotaxime, vancomycin, caffeine, ethanol, fluoride, and lactate. Ciprofloxacin demonstrated a particularly strong performance for the enrichment of Lachnospiraceae. Similarly, nanoparticles appear to better support the growth of Oscillospiraceae. Capsaicin showed no major differences compared to base medium. Overall, variations in phylogenetic diversity and composition of microbes were observed between the different conditions and original stool samples, indicating media modification and donor-dependent effects on taxonomic composition.

The Hellinger distance was used to assess the degree of similarity between media modifications and donors. A principal component

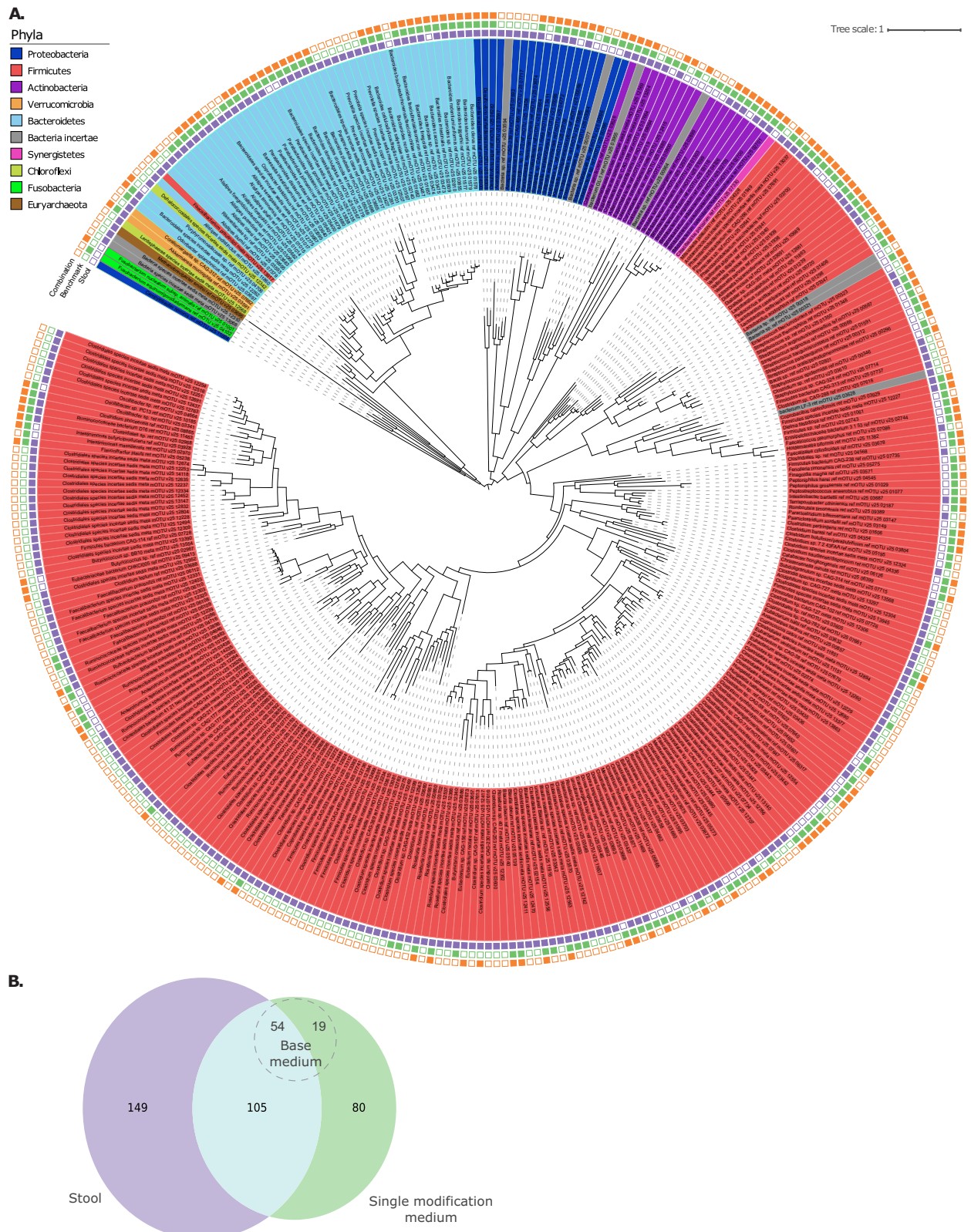

**Fig. 2 | Media modifications support the recovery of a larger taxonomic diversity. A** Phylogenetic tree of mOTUs recovered in the different samples. mOTUs are colored by phyla. Markers around the tree indicate if an mOTU was detected in stool samples (purple), single modification cultivations (green) or under combination of conditions (orange). **B** Venn diagram of the OTUs detected in stool and in single modification culture conditions. Source data are provided as a Source Data file.

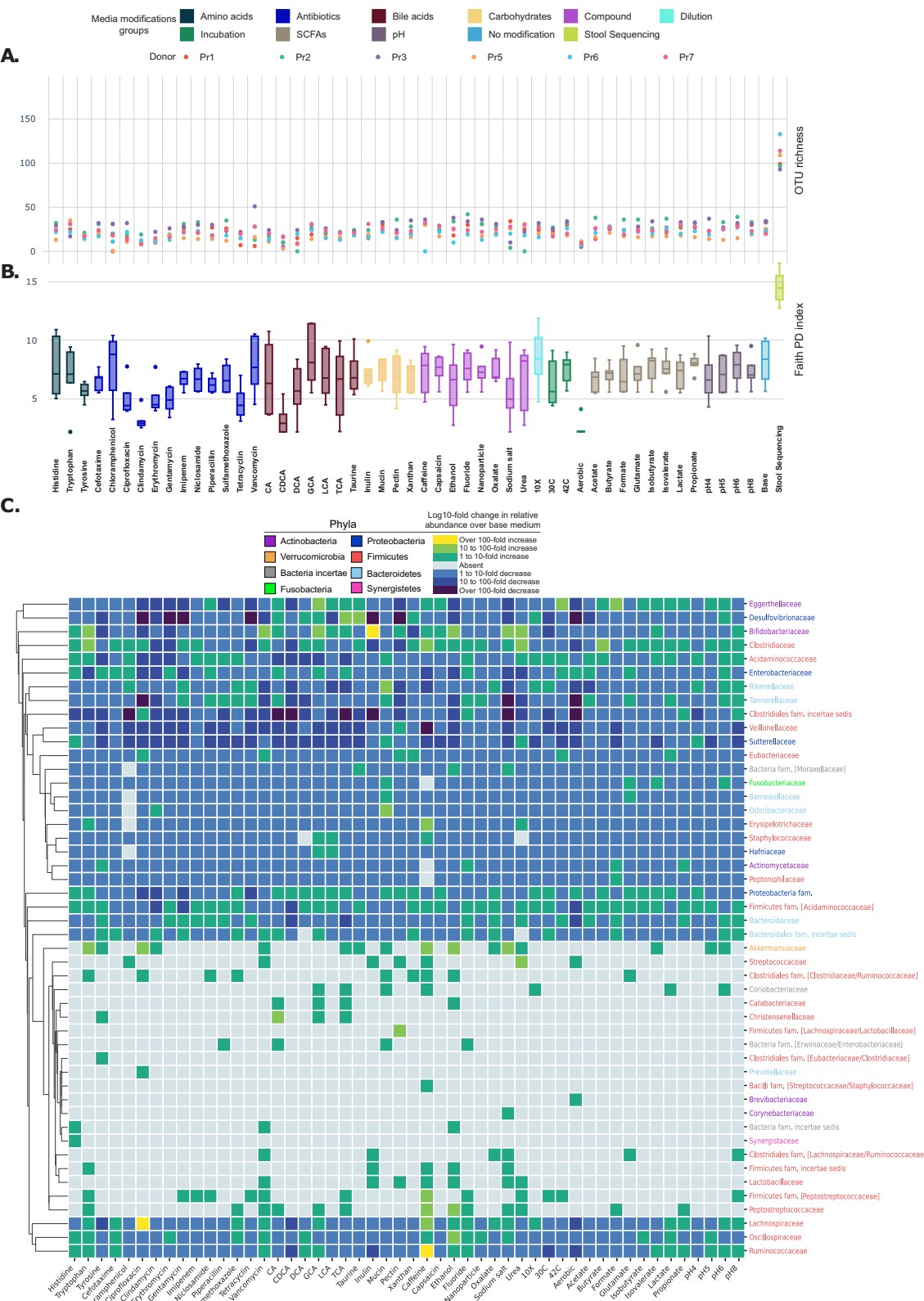

analysis (PCA) revealed that 40.8% of the variance could be explained by the first two axes, and highlighted that the initial composition of donors had a strong influence on the separation of samples. To further characterize the relationship between donors, we performed a Mantel test to assess the correlation between the patterns of community dissimilarities (as measured by Hellinger distances at the genus level) across donors (Fig. 4B). Positive correlations indicate that the overall

patterns of how media modifications affect community composition are similar between donors. This suggests that modifications causing large shifts in one donor's community tend to cause large shifts in others as well, although not necessarily involving the same genera. Finally, PERMANOVA analysis confirmed the significant impact of donor composition on genus-level microbiome composition ($F = 20.55$, $p < 0.001$) without affecting sample dispersion (PERMDISP:

**Fig. 3 | Taxonomic diversity is influenced by media modifications. A** Species richness across media modifications in the single modification set, colored by donor. Totals indicate the sum of unique mOTUs per donor in all modifications, including stool sequencing. **B** Faith's Phylogenetic Diversity (PD) index calculated for each individual sample across all medium modifications and donors. Each point represents the PD of a specific medium modification for a specific donor and is grouped by modification type and colored accordingly. Box plots summarize the distribution of PD values within each modification group ($n = 6$). Data are shown as medians with box plots illustrating the interquartile range (IQR; 25th–75th percentile). The horizontal line within each box marks the median, while whiskers extend to the smallest and largest values within 1.5 times the IQR. Points beyond the whiskers denote outliers, and individual points indicate raw data. No error bars are included, as this figure specifically uses a box plot format. **C** Heatmap of the mean Log10-fold change in relative abundance across donors for taxonomic families detected in single modification media modifications over GAM base. 0.001% was used as a baseline count for the calculations. Families are hierarchically clustered using Euclidean distance and the average linkage method to group families with similar abundance profiles across media modifications. Source data are provided as a Source Data file.

$F = 1.43$, $p = 0.204$, 999 permutations). These results indicate that the baseline donor composition substantially influences the microbiome composition across most medium modifications. However, modifications also demonstrated a significant effect on genus-level composition (PERMANOVA: $F = 2.42$, $p < 0.001$) without affecting sample dispersion (PERMDISP: $F = 1.01$, $p = 0.189$, 999 permutations), with some samples exhibiting taxonomic diversity beyond their initial donor composition (Fig. 4A, marked in lavender). To quantify the relative influence of donor composition and media modifications, we calculated the ratio of within-donor variance to between-modification variance for each medium modification (Fig. 4C and Methods). A lower ratio indicates that the variation in genus composition is more strongly influenced by the modification than by donor-specific differences, suggesting that the modification tends to drive microbial communities of different donors toward more similar compositions. Conversely, a higher ratio suggests that donor composition has a stronger influence, and the modification's effects vary among donors. Overall, the average ratio was below 0.1, indicating that the genus relative abundance variation due to modifications was, on average, ten times greater than the variation attributed to differences between donors. Some modifications, such as caffeine, resulted in different taxonomic diversity compared to the base medium (Fig. 3B, C) but had higher variance ratios (Fig. 4C), indicating a stronger influence from the initial donor composition. In contrast, other modifications, such as CDCA, had lower variance ratios, suggesting a more pronounced influence of the modification on taxonomy relative to donor influence and exhibited lower diversity.

In summary, while the initial taxonomic composition of donors was a major contributor to the final observed taxonomic composition, modifications also exerted a notable impact at the genus level. The extent of this effect varied across modifications, with some showing a more substantial influence on genus variation than the donor effect. However, modifications generally resulted in a more consistent taxonomic composition across different donors compared to the variation observed between different modifications.

## Growth in media modifications is associated with phylogeny

Previous work using strains from culture collections identified an association between phylogeny and medium composition preference[35]. We investigated whether growth in media modifications is associated with phylogeny by analyzing the similarity of growth profiles between taxa at different taxonomic levels (Fig. 5A). For each taxon, we constructed a growth profile based on its relative abundance across all media modifications. We then calculated the Hellinger distances between all possible pairs of taxa within each taxonomic level. The median pairwise Hellinger distance decreased from phylum to species level, indicating that more closely related taxa tend to have more similar growth profiles. This suggests that phylogenetic relatedness influences the response of taxa to different media modifications. Genus-level visualization of the growth profile distance in media modifications also confirmed this observation (Fig. 5B), highlighting a segregation of Bacteroides and genus belonging to the Proteobacteria phylum. This is consistent with the phylum level data (Supplementary Fig. 1), where *Bacteroides* and Proteobacteria are usually present and

dominant in most modifications, while other genera vary more broadly in occurrence. We observed that genera belonging to Firmicutes and Actinobacteria were all part of the same large cluster, revealing potential similarities in media preferences among taxa from these phyla. Finally, we calculated the number of media modifications in which each mOTU was detected per donor (Fig. 5C). Results indicated that a large proportion of mOTUs were detected in more than one media modification, with an average of 27.5 mOTUs only detected in one media modification. Leveraging this phylogenetic relationship could help determine what media modification would be the most efficient to enhance the cultivation and enrichment of specific taxa. Based on these observations, we hypothesized that combining some of these medium modifications could further increase the selective enrichment for specific taxa or microbial functions of interest.

## Selective enrichment of functional and taxonomic targets

To select combinations of media modifications for the enrichment of a desired microbial function or taxa, we devised a method that leveraged taxonomic and functional targets together with the data collected from the metagenomes of our single modification cultivations (Fig. 6A). We calculated and ranked combinations based on mean relative abundance and absolute count for targets across modifications for each donor and each combination. These metrics inform which combination had highest likelihood to enrich for chosen targets. A total of 32 combinations were generated to perform the targeted enrichment of a range of taxa and functions.

To demonstrate the efficacy of the method, we targeted *Collinsella aerofaciens*. This species is a prevalent member of the gut microbiota and has been found to be depleted in patients suffering from inflammatory bowel syndrome[50–52]. However, no selective medium exists in literature for the targeted cultivation of this species. Since it was present in only five of our single modification cultivations (Fig. 6C, green bars), we created a new medium that combined these modifications. This medium included pectin, inulin, 10X dilution base, lactate, and taurocholic acid. This combination of medium modifications yielded a significant improvement of 10 to 100-fold increase in the relative abundance of *Collinsella aerofaciens* (from 0.015% to 2%) over the single modification cultivation.

To further showcase versatility of our approach and its ability to enrich for microbial functions of interest, we sought to target microorganisms carrying synthetic pathways associated with dopamine and DOPAC synthesis (Fig. 6D). Dopamine and its metabolite DOPAC are neurotransmitters essential to human physiology, synthesized from tyrosine. A study performed in a large human cohort identified a strong correlation between the gut microbiota, DOPAC and mental quality of life[53], as well as a link to Parkinson's disease[54]. A combination of 4 media modifications (acetate, pectin, 10X dilution, pH6) was able to achieve 10 to 20-fold enrichment in dopamine and DOPAC pathways over the mean relative abundance in the base medium.

Interestingly, we observed mOTUs that were exclusively found in combinations of conditions that were not detected under single modification cultivations. This enabled the enrichment of 11 mOTUs that were not detected in single-component media, nor in the

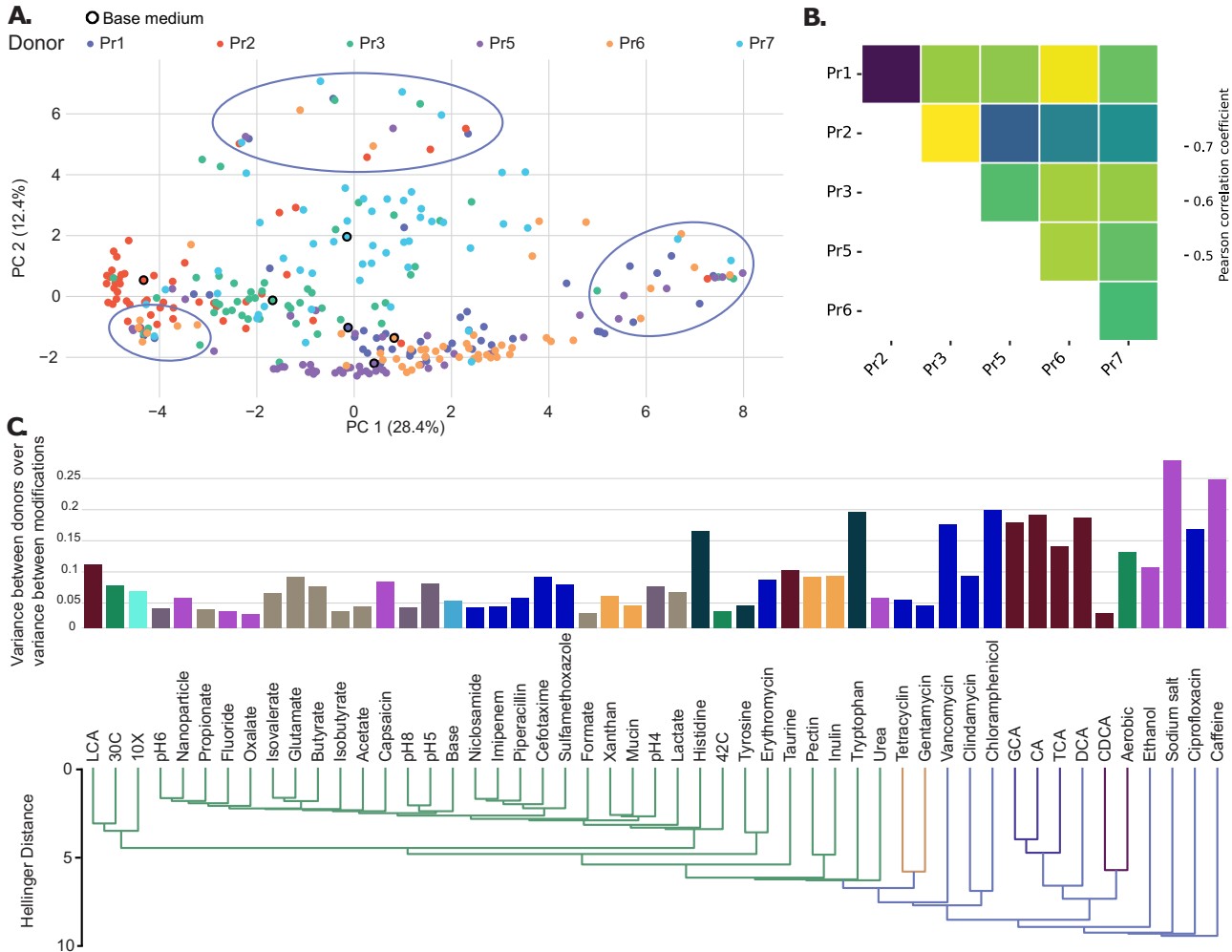

**Fig. 4 | Donor and modification effects on the taxonomic composition of samples. A** Principal component analysis of the Hellinger distance between modifications at genus level. Color corresponds to the different donors. Hand-drawn ovals indicate clusters that represent samples from multiple donors, positioned at the extremities of the plot. These ovals highlight areas where groups of samples share a similar biological response across different conditions, suggesting some shared effect between donors. The number of samples within each hand-drawn oval is arbitrary. **B** Heatmap of the Mantel test results between donor distance matrices using the Pearson R correlation coefficient (*p*-value < 0.01 for all correlation coefficients). The Hellinger distance between modifications at genus level is calculated for each donor, these distance matrices are leveraged in a mantel test to calculate correlations between donors. A positive correlation means that there is a correlation between distance matrices for a set of donors, and that conditions tend to enrich for the same genus in a particular medium. Pearson correlation calculations for the Mantel tests were performed using a two-sided approach. **C** Ratio of the variance within donors to the variance between modifications for each condition. Lower ratio values indicate that the genus relative abundance variation within donors is small compared to the variation between modifications. Higher ratio values suggest that donor composition has a stronger influence on genus variation within modifications. Media modifications are colored by groups and ordered according to their clustering using the Hellinger distance on the mean relative abundance (average linkage method, cophenetic correlation coefficient = 0.943). Modifications further away from Base support larger differences in taxonomic composition. Source data are provided as a Source Data file.

combinations (Fig. 6B). To quantify these differences, the proportion of species and functions present in the various modifications that make up the combinations was calculated for each sample (Fig. 6E). The result revealed that in combinations composed of two modifications, only 55% of the species detected in the combinations are present in the two modifications in our single modification cultivations. For 5–6 modification combinations, around 30% of mOTUs are present in all the conditions. These results indicate that a majority of the species present in the combinations were not able to grow in media containing a single modification.

We also observed certain combinations that outperformed conventional media. A combination of lactate, mucin, fluoride, nanoparticle and 10X dilution showed markedly increased enrichment of *Lactobacillus* and *Bifidobacteria* compared to the gold-standard MRS medium (Fig. 6F). Contrary to single media modifications, the PERMANOVA between combinations grouped by donors did not show a

significant effect of donors on the final genus-level composition (F = 1.38, *p*-value = 0.108, permutations = 999). However, the effect of combinations on genus-level composition in the samples remained significant (F = 2.30, *p*-value < 0.001), with no observed effect on the dispersion between combinations (PERMDISP F = 1.08, *p*-value = 0.162, permutations = 999). These results indicated that combinations of modifications are impacted less by composition of the original donor sample, compared to the modifications alone, which could be a result of the greater selective pressure applied on the microbial communities, thus more systematically enriching for a common reduced set of taxa across donors. Furthermore, it might suggest that perceived differences in some community structures may be a consequence of differences in relative abundance, rather than the presence/absence of a specific strain.

Finally, the average number of species detected in combination cultivations was similar to unique modifications in the single

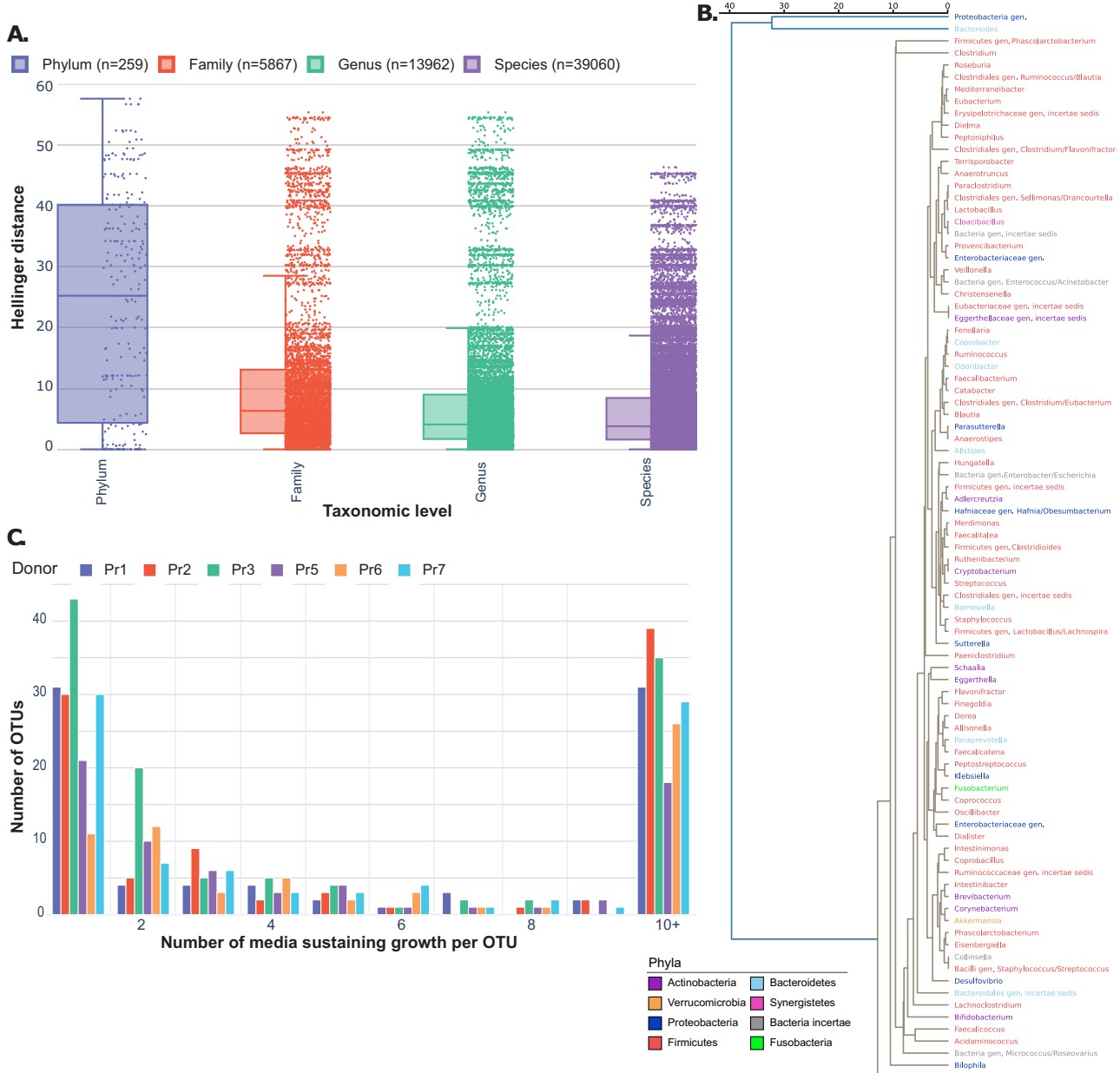

**Fig. 5 | Growth profile distance between modifications is associated with phylogeny. A** Distribution of pairwise Hellinger distances between taxa at different taxonomic levels (Phylum, Family, Genus, Species). Each point represents the Hellinger distance between the growth profiles of a pair of taxa at the specified taxonomic level. The decreasing median distances from phylum to species level indicate that more closely related taxa have more similar growth profiles, suggesting a phylogenetic influence on growth preferences. Data are presented as box plots to illustrate the distribution of growth profile differences based on Hellinger distance across various taxonomic levels. The central horizontal line within each box represents the median, while the box spans the interquartile range (IQR;

25th–75th percentile). Whiskers extend to the smallest and largest values within 1.5 times the IQR, with points beyond the whiskers indicating outliers. Individual data points are shown alongside the box plots to provide additional context on the variation in growth profiles. No error bars are included, as the figure emphasizes the box plot representation. **B** Dendrogram of the growth profiles' distance between genus based on Hellinger distance (average linkage method, cophenetic correlation coefficient = 0.933). Genus is colored by phylum. **C** Number of media modifications sustaining growth per mOTU. mOTU count is calculated by donor, indicated by colored bars. Source data are provided as a Source Data file.

modification set (21.8 and 21.3, respectively). This result is consistent with the number of mOTUs detected between single modification and combinations, suggesting a potential for further enrichment with additional modifications.

## Discussion

In this study, we developed a streamlined approach for cultivation and targeted enrichment of gut microbes and functional capabilities of interest. Using a base medium with 50 different modifications, we

significantly expanded the taxa cultivated compared to the base medium alone. Leveraging whole metagenome sequencing, we identified modifications of culture media that enrich for particular taxa, such as the *Lachnospiraceae*, *Oscillospiraceae* and *Ruminococcaceae* families, which are often associated with healthier subjects[19]. Furthermore, we observed that taxonomic composition of the communities detected on the plates was significantly influenced by the donor, with some media modification effects at genus level translating across donors.

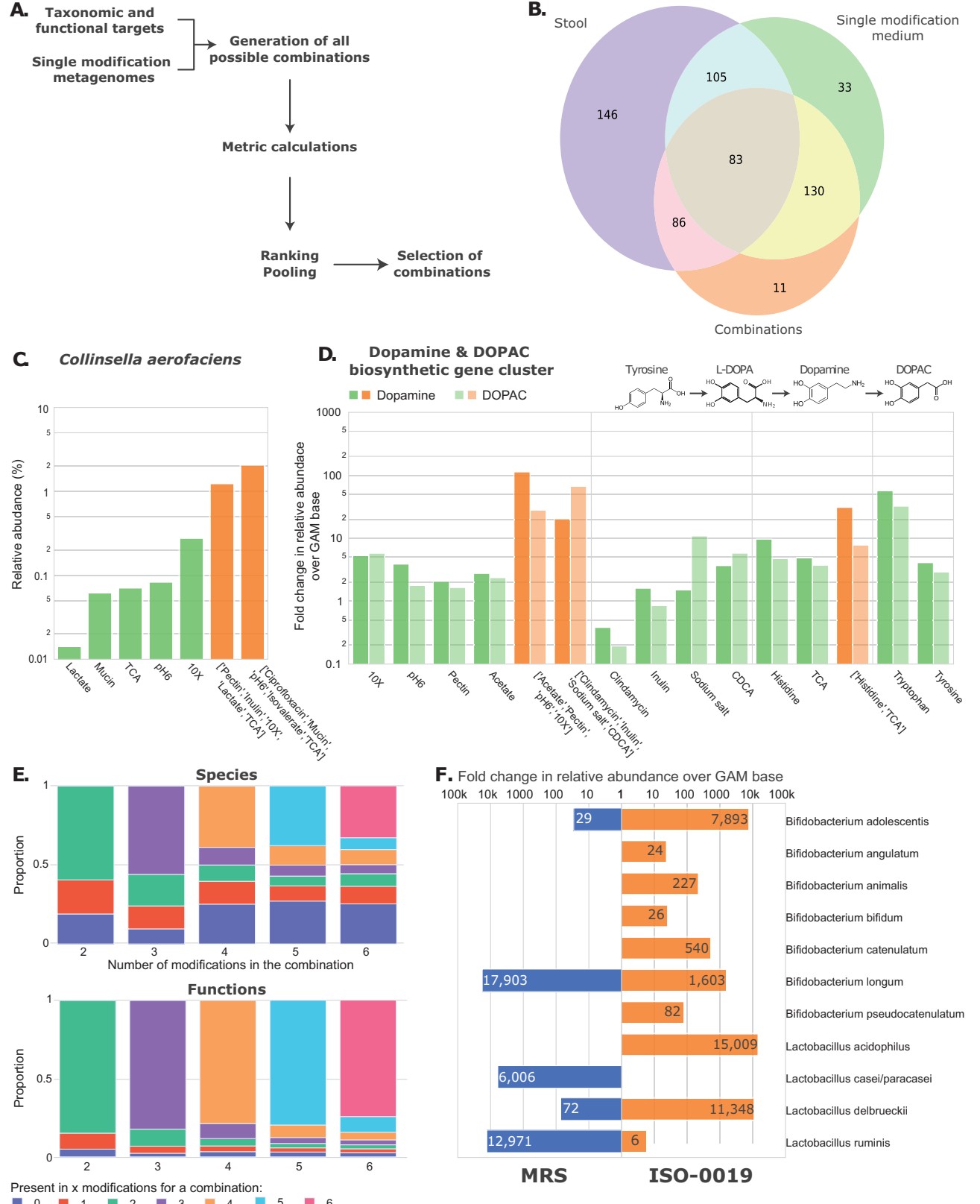

In the single-modification cultivations, the phylogenetic distance appeared to be associated with the growth profiles of the different taxa. Informed by our initial single-modification cultivations, we designed combinations of conditions aimed at increasing the targeted enrichment for desirable taxa and functions, such as *Collinsella aerofaciens*, or biochemical pathways involved in the metabolism of dopamine and DOPAC. Our results highlight the potential for the recovery of strains with functional capabilities that have not been associated with specific taxa. Finally, our combinations of conditions uncovered a culture media composition with improved recovery of *Lactobacillus* and *Bifidobacterium* species compared to MRS, even for low abundance or rare species[55].

**Fig. 6 | Combinations of conditions further improve the targeted enrichment of taxa and functions of interest. A** Overview of selection process of modification combinations. Depending on the targets (microbial functions or taxa), different metrics can be calculated, in order to achieve the maximum likelihood for enrichment, based on the co-occurrence of targets in the different modifications (see "Methods"). **B** Venn diagram of the OTUs detected in stool, single modification, and combinations. **C** Bar plot for the enrichment in relative abundance (%) for *Collinsella aerofaciens*. Single modification cultivations are colored in green and combinations in orange. **D** Bar plot for the fold change enrichment over Base for genes associated with dopamine and DOPAC biosynthetic gene cluster, based on functional assignment to KEGG orthologous groups for all samples. **E** Proportion of species or functions present in different modifications composing the combinations, for different numbers of modifications per combination. Each bar represents the proportion of species or functions detected in the specified number of modifications. For example, in combinations composed of 4 modifications, close to 40% (orange) of the species are detected in all 4 modifications, while around 30% (dark blue) are not detected in any single modification composing the combinations. **F** Fold change enrichment over Base in logarithmic scale for *Bifidobacterium* and *Lactobacillus* species in MRS and ISO-0019. Source data are provided as a Source Data file.

Contrary to common culturomics assumptions, we observed that a single modification to the base medium yielded significant recovery of microbial species, challenging the need for studies that use extensive media panels, as previously demonstrated[56]. Altogether, it suggests that modulating the composition of a single medium, or using a reduced set of media that strongly differs by their base composition, may be more efficient than the use of a large media panel in culturomics. Furthermore, limiting variations of culture media will foster more comparable outcomes and lead to improvements in study design.

Leveraging media modifications to drive selection and microbial interactions presents both advantages and challenges. Indeed, gut microbial communities are dominated by ecological forces such as exploitation, competition, and amensalism[57]. Simultaneously, culture-independent studies have indicated that a large number of taxa in the human gut possess auxotrophies and reduced genomes[16]. As a consequence, such strains may require more defined growth characteristics, such as our observation of unique OTUs that were only detected when conditions were combined. This points to challenges in rationally designing combinations based on single-modification data alone. Moreover, in certain media modifications, lack of detection of some species does not necessarily indicate lack of growth and could stem from multiple factors, such as biases during the sample preparation and sequencing, the absence of the species in the growth condition, a low relative abundance in the samples, or the impossibility to grow and compete against other microbes in the presence of the modification.

Using both solid and liquid culture systems could enhance throughput, reduce costs, and facilitate independent taxonomic studies. While this study focuses on modification effects on taxa growth, future studies may implement this strategy to investigate the influence of modification on the occurrence of, for example, pathogenicity factors, phages, biosynthetic gene clusters or CAZymes. Furthermore, the methodology presented in our manuscript, when combined with automation, could significantly enhance the recovery and isolation of less abundant taxa, advancing culturomics studies and help building large culture collections.

The analyses presented here was performed on unassembled metagenomic data, and leveraging metagenome-assembled genomes may provide more precise tools to track the co-occurrence of functions and taxa between modifications. Furthermore, modeling applications, such as decision trees or support vector machines could help determine the importance of features, albeit modifications, function or taxa depending on the experimental objectives. Finally, deep learning and graph neural networks could provide a means to cohesively connect information such as donors, OTUs, function and media together, enabling the prediction of suitable combinations for a target taxon that are based on the initial composition of the donor.

In summary, this study presents a unique culture-based method that uses widely available materials to enrich in a targeted manner for the growth of microbes with desired phylogenetic or functional features. Our use of a modified single base medium introduces a conceptual approach towards ongoing culturomics efforts, and we provide a resource to allow further exploration of different microbial communities and conditions that facilitate targeted isolation of microbes contained within them. This work will help guide more streamlined and rationally designed strategies to enable the cultivation and study of previously inaccessible bacteria, particularly those with medical relevance or translational potential, furthering microbiome research and the development of microbiome-based therapeutic interventions.

## Methods

### Ethical statement
The study was pre-assessed by De Videnskabsetiske Komiteer for Region Hovedstaden under case reference number 22013134. The committee concluded that the project was not a health science research project and therefore did not require ethical approval.

The study adhered to the ethical guidelines of the Danish Technical University and complied with all applicable EU regulations. Data collection and processing were conducted in full accordance with the GDPR. Written informed consent was obtained from all participants, explicitly permitting the use of their samples for research purposes. Participation was entirely voluntary, with no financial or other forms of compensation provided. Sample were anonymized, this decision aligns with the committee's guidelines, which state that research involving only completely anonymous human biological material collected according to relevant legislation does not require reporting.

### Sample collection and preparation
Participants were selected to be considered healthy individual and, therefore, to fulfill the following self-declared inclusion criteria prior to donation: males or females, aged 20-60, not following a particular diet regimen and without underlying or chronic condition, not having undergone antibiotic, antiviral, antifungal or antiparasitic treatment over the 12 months prior donation, not suffering of gastrointestinal pain, discomfort, intolerance or been subject to a surgical procedure.

Participants were requested to self-sample using a kit composed of a container containing antioxidant solution, a GasPak anaerobic pouch and nitrile gloves. Once the donation was completed, participants anonymously deposited the sample in a dedicated box stored at 4 °C for subsequent processing over the next hour the same day to preserve sample viability. Fresh fecal samples were obtained from six consenting healthy adult participants and were placed in anaerobic conditions in a Don Whitley A95 anaerobic chamber at ambient temperature (5% Hydrogen, 20% Carbon dioxide, 75% Nitrogen). The samples were then prepared for cryopreservation[58]. Briefly, each fecal sample was homogenized (0.1 g stool per ml PBS) in pre-reduced PBS containing an antioxidant mix (0.1% ascorbic acid, 0.01% glutathione, and 0.4% uric acid). Multiple cryostocks were made for each sample using a final concentration of 20% glycerol. Finally, a layer of mineral oil was added to each tube to prevent gas exchange with the cryostock. All the cryotubes were stored at −80 °C. While direct processing and use of the stool sample would likely increase the recovery and diversity obtained, performing the cultivation from cryostock enable the collection of multiple stool samples over time before the start of a campaign, and decouple the cultivation experiment from the collection process, making it more broadly applicable.

## Microbiota culture

Prior to the media preparation, all plates and containers received a single identifier and were labeled with barcodes to ensure tracking and quality control. The base medium (GAM-STD) used for all the plates is a modified version of the Gifu Anaerobic Medium (GAM). GAM-STD contains, per liter of medium, 59 g of medium powder (HyServe), 10 mg of vitamin K1, 10 mg of Hemin, 13.5 g of agar, 1 g of ascorbic acid, 0.1 g of glutathione and 4 g of uric acid. The final pH is adjusted at 7.3 before autoclaving. The complete list of media modifications, along with the media identifier and final concentration of the additives used in each variant, can be found in Table 1 of the Supplementary materials. All the additives can be sourced from Sigma-Aldrich.

For the preparation, all the media components were pre-reduced and anaerobically sterilized (PRAS) to minimize oxidation damage and preserve the media properties. Briefly, all powders were incubated in the anaerobic chamber for 24H. The necessary volume of MilliQ water was then boiled and flushed with N2 to remove the dissolved oxygen. Powders were dissolved anaerobically according to manufacturer's recommendations and sealed in serum bottles for autoclaving at 115 °C for 20 min. The base medium (GAM) was autoclaved separately from the selective agents and the agar to avoid the formation of radical oxygen species. Heat sensitive components were filter sterilized using a 0.2 μm filter. All the media components were then stored at 4 °C, unless stated otherwise by the manufacturer, until the plate pouring for less than 72H. Nunc OmniTray single well plates (ThermoFischer Scientific) were selected for the experiment to maximize cultivation surface while keeping the total volume occupied by plates manageable. 35 mL of media was poured in each plate aerobically under sterile conditions using a peristaltic pump to ensure consistency. The plates were prepared and placed in anaerobic conditions 48H prior to inoculation, along with PBS and all other materials, to allow for reduction and to control for batch contamination.

Cryostocks less than 6 months old for each of the 6 donors were thawed one-hour prior inoculation. All the inoculation process was performed in anaerobic conditions in a Don Whitley A95 anaerobic chamber at ambient temperature. Each stools sample from the cryostocks was serially diluted using sterile reduced PBS to achieve the optimal cell density. 2 serial dilutions (100X and 1000X) per stool sample were selected to inoculate (100 uL) the chosen media prepared earlier and spread with sterile glass beads. Growth is influenced by the initial cryostock cell density and the impact of preservation. Therefore, dilutions need to be tested on base medium prior to the cultivation campaign (mGAM without modification) to select the most suitable dilution. All the inoculated plates were incubated anaerobically (except for the aerobic selective condition) at 37 °C (besides the 30 °C and 42 °C conditions) in jars containing AnaeroGen anaerobic pouches (Oxoid).

## Sample preparation for sequencing

7 days after inoculation, all the plates were imaged and cultures were manually scraped on the entire plate surface to collect all colonies into 2 mL Eppendorf tubes, using 2 mL PBS to facilitate the colony detachment. Only one dilution plate out of the two created for each sample was scraped, leaving a single dilution per sample to prepare for sequencing. The scraped plate was selected to achieve the maximum cell density that can be recovered without obtaining a lawn. The scrapes were then transferred from Eppendorf tubes to 96 2 mL deep well plates using an Opentron OT2 liquid handler to facilitate the subsequent analysis. Metagenomic DNA extraction of the scrapes was performed using the DNeasy UltraClean 96 Microbial kit (Qiagen) according to manufacturer's instructions. The extracts were stored at −20C for preservation until Quality Control and concentration measurement, performed using a Quant-iT PicoGreen dsDNA kit (ThermoFischer Scientific) and measured on a Synergy plate reader (Agilent).

Extraction of the stool sample metagenomic DNA was performed directly on the initial cryovial used for the dilution and plating. The DNA was extracted using a DNeasy PowerSoil kit (Qiagen) according to the manufacturer's instructions. The pure DNA was then stored at −20C for preservation until it is processed with the DNA from the plate scrapes for Quality Control. The quality control was performed on all DNA extracts using a NanoDrop 2000 (ThermoFischer Scientific) to verify purity and a QuBit (ThermoFischer Scientific) for fluorometric quantification.

The Whole Metagenomic Sequencing library was prepared using the plexWell 384 Library Preparation Kit, as it is designed to perform high throughput multiplexed library preparation for up to 384 samples, according to manufacturer's instructions. The kit is optimized for a 10 ng purified dsDNA input. The quality control of the prepared library was analyzed on NanoDrop 2000 and QuBit to measure concentration and purity, as well as a Bioanalyzer DNA 1000 kit (Agilent) to verify the insert length distribution.

## Sequencing and data analysis

Whole metagenome shotgun sequencing was performed on the prepared library by a third-party service provider, BGI sequencing, on a single lane from a Novaseq 6000. The pair-end $2 \times 150$ bp reads were observed at a depth of $13,287,651 \pm 2,279,805$ (mean ± SD) reads for stool samples and $4,569,357 \pm 3,411,614$ (mean ± SD) for plate scrapes. No rarefaction of reads was performed. Sequencing data were processed using the NG-Metaprofiler package[59] to obtain taxonomic profiles, based on the mOTUs version 2.5 database[60], and functional assignment to KEGG orthologous groups (KO). Functional assignment to KO groups was used to design media for functional targets while taxonomic profiles were used for the design of media towards specific taxa such as *Collinsella aerofaciens*, following the process described in Fig. 6A. Briefly, the design of targeted media was done in 4 steps (Supplementary Fig. 2). First, all possible combinations of media were generated for the conditions used in the single modification cultivations, up to 6 conditions per combination, amounting to 10.9 M combinations. Then, metrics were calculated to score what could be the most likely combinations to yield an enrichment for the target:

1. Mean relative abundance per OTU/Function ($MRA_{target}$): The mean relative abundance of the target taxon or function across the modifications included in the combination was calculated using the formula (1) $MRA_{target} = 1/n \sum_{i=1}^{n} RA_{target,i}$, where $n$ is the number of modifications in the combination and $RA_{target,i}$ is the relative abundance of the target in modification $i$.

2. Total Mean Relative Abundance of All OTUs/Functions in the Combination ($MRA_{total}$): The total mean relative abundance of all operational taxonomic units (OTUs) or functions detected across the modifications in the combination was calculated as (2) $MRA_{total} = 1/n \sum_{j=1}^{m} MRA_{OTU_j}$ where $m$ is the total number of OTUs/functions detected across all modifications in the combination and $MRA_{OTU_j}$ is the mean relative abundance of OTU/function $j$.

3. Ratio of Target Mean Relative Abundance to Total Mean Relative Abundance ($R_{target/total}$): This ratio indicates the proportion of the target's mean relative abundance relative to the total mean relative abundance of all OTUs/functions in the combination and is represented as (3) $R_{target/total} = MRA_{target}/MRA_{total}$.

Finally, combinations were ranked for each target based on highest $R_{target/total}$ > lowest total number of OTUs/function $m$ > highest mean relative abundance of the target $MRA_{target}$. The top 20 modifications for all the combinations for a target were retained. Multiple visualizations were generated in a dashboard to assist in choosing the combinations that could be valuable to test: co-occurrence network of the most present modifications in the best combinations across donors, heatmap of the best modifications relative abundance per

donor and the mean relative abundance of the target relative to the number of OTUs/functions in the combination. An example of the code for this process is shared on github at https://github.com/Jerarm/Metagenome-guided-culturomics-for-the-targeted-enrichment-of-gut-microbes.

Data handling, calculations and statistical modeling were performed using Python 3.8 and the packages Numpy 1.22, Pandas 1.3, Scipy 1.7 and Scikit-learn 1.0. Ecological metrics such as alpha diversity, beta diversity, phylogenetic tree, PERMDISP, PERMANOVA and ANCOM were calculated using Scikit-bio 0.5. Phylogenetic tree was visualized with iTOL[61]. All plots were generated with the python packages Matplotlib 3.5, Seaborn 0.11 and Plotly 5.5. PCAs were generated using Scikit-learn. Figures were made using Adobe Illustrator CC.

The Hellinger distance was selected for distance-based calculations, and represents the Euclidean distance applied to the square root of relative abundance data. Hellinger transformation puts the species abundances on the relative scale, thus reducing the total abundance per sample discrepancy that can occur due to sampling differences, and square-rooting lowers the dominant species' importance. Hellinger distance is also asymmetrical (not influenced by double zeros) and has an upper limit of sqrt{2}, which makes it a suitable method for ecological data with many zeros. Finally, the Hellinger distance was chosen in the study because it offers a great compromise between linearity and resolution, thus making it suitable for linear ordination. Conforming to Euclidean space allows the use of distance-based calculation and tools developed for dimensionality reduction such as PCA, UMAP, and other clustering algorithms. Nevertheless, other distances were used in the exploratory phase, such as Bray-Curtis, with results being robust to the choice of the distance metric.

Faith's Phylogenetic Diversity (PD) was calculated for each individual sample using the observed mOTUs and their phylogenetic relationships. A comprehensive phylogenetic tree was constructed using all mOTUs detected across samples, based on the alignment of conserved marker genes from the mOTU database. For each sample, PD was calculated by summing the branch lengths connecting the taxa present in that specific sample using the 'faith_pd' function from the scikit-bio 0.5 library.

To assess the relative influence of donors and medium modifications on genus-level composition, we calculated the ratio of within-donor variance to between-modification variance (Fig. 4C). For each modification, we calculated the within-group variance by measuring the variance of genus abundances within each donor group. The between-group variance was calculated by evaluating the variance of the average genus abundances across different donors for each modification. The variance ratio for each modification was obtained by dividing the total within-group variance by the total between-group variance, with a lower ratio indicating a stronger influence of modifications. PERMANOVA was used to test for significant differences in community composition between donors and modifications, comparing the variance within and between groups. PERMDISP assessed the homogeneity of variances within groups. Together with the variance ratio, these methods provide a comprehensive understanding of the factors influencing microbiome composition, identifying both the significance and the relative impact of modifications versus donor effects.

Data tracking of all samples from sampling to sequencing analysis was handled via barcoding and recorded in Benchling. No statistical or randomization methods were used to design the experiments. The investigators were not blinded to sample allocation during the experiments or the analysis.

### Statistics and reproducibility
No statistical method was used to predetermine sample size. No data were excluded from the analyses. The experiments were not randomized. The Investigators were blinded to allocation during experiments and outcome assessment as sample donation was performed anonymously.

### Inclusion and ethics statement
This study adhered to the principles of inclusion and ethical responsibility. All collaborators who met the authorship criteria required by Nature Portfolio journals have been included as authors, as their participation was essential for the study's design, implementation and analysis. Contributions of the authors were recognized according to established authorship criteria, with additional contributors acknowledged where appropriate.

### Reporting summary
Further information on research design is available in the Nature Portfolio Reporting Summary linked to this article.

## Data availability
The sequencing data generated in this study has been deposited in SRA database under the BioProject accession number: PRJNA1077691. Source data are provided with this paper.

## Code availability
Metadata for the sequencing data and code related to the manuscript can be found on GitHub at https://github.com/Jerarm/Metagenome-guided-culturomics-for-the-targeted-enrichment-of-gut-microbes[62].

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

## Acknowledgements

This study was funded by the European Union's Horizon 2020 research and innovation program under the Marie Skłodowska-Curie grant agreement No. 813781 (J.A. and M.O.A.S.), the Novo Nordisk Foundation under the grant agreement NNF20CC0035580 and the The Novo Nordisk Foundation, Challenge programme, CaMiT under grant agreement: NNF17CO0028232 (J.A., S.S.L., T.H.V., R.V.U., and M.O.A.S.). S.S.L. acknowledges support from the European Molecular Biology Organisation (no. ALTF 137-2018) and the National Health and Medical Research Council of Australia (no. GNT1166180). The authors also thank the anonymous donors, as well as the members from Morten Sommer's group and Carolyn Bayer for their assistance with this study.

## Author contributions

J.A. and S.S.L. conceived the project. J.A. planned and performed the experiments. S.S.L. processed the raw sequencing data. J.A. analyzed the processed sequencing data. J.A. wrote the final manuscript in consultation with S.S.L., T.H.V., R.V.U., and M.O.A.S. M.O.A.S. supervised the study.

## Competing interests

The authors declare no competing interests.
