## [Transparent Peer Review file · Nature Communications]

Metagenome-guided culturomics for the targeted enrichment of gut microbes

Corresponding Author: Professor Morten Sommer

Version 0:

Reviewer comments:

Reviewer #1

(Remarks to the Author)

Thank you for giving me the opportunity to review this article. Over the past 12 years, culturomics has become a key technique for studying the digestive microbiota, thanks in particular to the prospects it offers for modulating the microbiota. The authors propose a very interesting work.

I find this manuscript particularly well written and clear.

I have several comments:

- 1) I suggest that the authors describe their supplementary material culture protocols even more exhaustively, so that the scientific community can reuse these techniques.
- 2) Readers may be disappointed by the fact that culture guides metagenomics but strains are not obtained. I think the authors need to discuss this limitation more precisely and could present it as a preliminary study. Indeed, only pure culture can enable a therapeutic approach.
- 3) A recent article (Nat Biotechnol . 2023 Oct;41(10):1424-1433. doi: 10.1038/s41587-023-01674-2. Epub 2023 Feb 20.) discusses the automation of culturomics. How do the authors position their work in relation to these perspectives?

Reviewer #2

(Remarks to the Author)

Summary

The authors seek to develop generalizable approaches to improving the cultivation of gut microbes. They apply a panel of enrichment conditions (modifications), individually and in selected combinations, to stool samples collected from six healthy adult volunteers. Stools and enrichments were surveyed using metagenomic sequencing. Compositional variation among enrichments (scraped plates) was explored with respect to donor (inoculum) community composition; the type(s) of modification(s) made to the base medium; and bacterial taxonomy. The strengths of the work include the systematic exploration of a wide range of individual modifications while holding all else constant, and the inclusion and consideration of different donor (inoculum) communities. Several areas where the work could be improved would be (1) a clearer, more specific, more detailed statement of the problem or question addressed by the research, and the bigger-picture goal; (2) clearer and more thoughtful communication of results and rationale for selecting or singling out individual modifications, taxa, and functions for focused study; and (3) supplying additional methodological details. The following comments are intended to help the authors improve the communication of their work.

Comments

Title:

I don't think metagenomics "guided" the culturomics; rather, it was metagenomics-based? Metagenomics was the read-out. Also, I don't feel the methods were really that "targeted", at least not from an a priori design perspective that was communicated.

Methods:

Line 424-425: Two serial dilutions were used to inoculate each medium. What became of the two dilutions? There's no other mention of them.

Line 429-434: Understanding the work depends a lot on understanding these Methods. Please address the following: (1) The presence of confluent colonies resulted in the elimination of the whole plate, or simply avoidance of the confluent colonies while scraping other regions of the plate? And what's the problem with confluent colonies, anyway, given that isolation was (seemingly) not a goal of this study? (2) Was the entirety of each plate scraped into a single 2 mL tube, along with the 2 mL of PBS used to facilitate colony detachment? Or was each colony of each plate placed into a separate tube? This is relevant to understanding the meaning of "relative abundance" in the context of the plate scrapes. (3) What was the purpose of imaging if all biomass from each plate was scraped into a single tube? Or was each colony tracked?

Line 459: Why was the Hellinger distance selected here? Were the results robust to choice of distance metric? Also, what Methods (packages) were used for PCA (PCoA?); PERMDISP & PERMANOVA.

Line 466: The BioProject accession does not exist in a public-facing format. If the data are pending release, then a reviewer link should be made available. The authors are also encouraged to plan to make their analysis code available.

Results:

Throughout: The following terms are used: stool, scrapes, plates; base, baseline, benchmark; modifications (unique or individual); and combinations. It gets confusing. Please define your terms and use them consistently.

Line 131: If a given mOTU was detected in the original stool of 6 out of 6 study participants, but was detected in only one participant's collection of benchmark cultivations, how many times was it considered absent?

Line 137 "that is inaccessible": Your result is that cultivation is inaccessible to cultivation-independent methods? This seems obvious. Maybe you mean that the diversity was not accessed, as in:

"Our cultivation approach enriched for diverse taxa that were undetected at the depth of sequencing (arbitrarily) used in our cultivation-independent surveys."

Line 139: On the one hand, it's perhaps not surprising that different culture conditions enrich for different taxa. On the other, it's certainly interesting to interrogate how single modifications might shape or shift the enrichment landscape. I wonder if additional visualization might help develop intuition about this selective landscape. For example, approaches like RDA or CCA which could reveal modifications (or groups of modifications) that most strongly associate with enrichment or depletion of particular taxa (or groups of taxa) across multiple participants.

Line 149: Start of sentence (or more?) is missing. Feels like maybe a whole paragraph has been accidentally deleted.

Figure 3B: Boxplots missing for last two categories at right? Also, the color difference between amino acids and antibiotics is undistinguishable to me, here and throughout.

Figure 3C: Would love to see an additional panel showing participant- and species- resolved data for the three modification-family pairs with big increase (the yellow cells in panel C).

Line 164: Not sure this statement follows from Figure 3C. At least it's not obvious to me.

Line 180-181: Would it be more accurate to say they've been associated with states of health?

Line 185 "significant": Please state the details of any statistical tests.

Line 199-200: The percentage of variation explained says little about the source of the variation. If you observe clustering by participant, meaning that media modifications don't completely override that signal, then one might say that donor composition is reasonably stable to most selective regimes. (In some ways, it must be: there's no alternative source of microbes – none other than the unsampled tail of the rank-abundance curve.)

Line 200-201 and Figure 4A: What is called "baseline" – the stool or the base media? Also, unclear whether "baseline" samples even appear in Figure 4A (given the caption). And finally, because the (Figure 4A) plot regions representing the highest values of PC1 and PC2 contain samples (ostensibly from certain modifications) from numerous participants, it seems unlikely that baseline variation among donors has the strongest influence on the compositional variation depicted on this plot. I think what you're trying to depict is that most modifications reflect baseline clustering of samples by participant (in other words, the modifications cluster near the stool or base); while just a few modifications (the extremes noted here) reflect clustering by modification. Would it be possible to highlight each participant's unaltered stool sample with a different shape or outline? Or perhaps provide a supplement in which the samples are colored by the few modifications you wish to highlight? (Or label the ovals so the reader might understand which modifications are highlighted?)

Line 201-202 and Figure 4B: I understand what a Mantel test is, but I don't understand what you've done here. Is this

showing the degree to which pairs of participants had similar responses to the cultivation regimes, in terms of which conditions pushed the composition more or less away from their own baseline? For example, participant 2's within-participant distances are most highly correlated with participant 3's within-participant distances?

Line 207: To which comparison does this p-value apply? Is this a PERMANOVA of "stool" (or "base") versus each "mod", or other? Please take greater care to explain your results.

Line 209: I do not see how the statement on this line is supported by Figures 3B and C, because you have not mentioned which modifications or samples these are (at the periphery of Figure 4A).

Line 211: Which lavender region ("cluster")? There are three. Are these confidence ellipses? Or were they drawn by hand? Why mention (whichever one is) CDCA, and not the others?

Figure 4C and lines 211-213: Figure 4C is not mentioned in the text. The sentence ending on line 213 might refer to Figure 4C, but the sentence is too confusing (to me) to reconcile with the figure.

Line 231: It stands to reason that phylogenetic relatedness predicts ecological similarity.

Line 236 (Figure 5B), highlighting a segregation: I don't really see it; not sure I understand.

Line 257: taxon

Line 266: *Collinsella aerofaciens* was present in five modifications ... all from the same participant?

Line 269: I'm not sure I understand the goal of the approach, or the interpreted meaning of relative abundance in this enrichment context. Is the goal here to reduce the number of colonies one has to sift through before hitting this species (on a plate that enriches for many)? Is the goal to obtain an isolate from an individual patient?

Line 276: Why expressed as fold-change whereas *Collinsella aerofaciens* data were expressed as relative abundance? Prefer the latter. Ditto for Figure 6E.

Lines 298-300: Yes, I think that's right. In other words, there's probably a lot more "non-detection" than true absence. I think this is generally recognized.

Reviewer #3

(Remarks to the Author)

This manuscript presents the results of a set of "culturomics" experiments in which microbes were cultivated from stool samples using a large number of variations on a standard medium, then characterized via shotgun metagenome sequencing to analyze the impacts on overall diversity as well as changes in the retrieval of individual species. The investigators found that even small changes in media composition resulted in changes in the diversity and composition of the organisms cultivated from multiple stool donors. They also found that combining modifications could result in further changes in cultivable communities, which were not always predictable based on the results of the single modifications.

This is a nice study that could be of interest to a range of scientists, particularly those studying gut communities but also those studying a range of microbiomes. However in many cases the results have only been minimally explored and the outcomes aren't always clear. A few overall points:

The distinction between the "benchmark" experiments and the "combinations" is never clearly explained and has to be inferred from the descriptions, which is particularly hard when the combination results are included in Figure 2 but the combinations are not described in the narrative until line 256, well into the results section. I understand the flow of the narrative, but I think the specific meaning of "benchmark conditions" in this manuscript needs to be provided when first mentioned at line 126, even if the full explanation of the combinations is left until later.

The title describes "metagenome-guided culturomics" but it appears that metagenomes were used primarily as a phylogenetic readout. The use of metagenomic information to guide the combination cultivations had exciting promise, but instead nearly all analyses relied solely on extracted 16S fragments (mOTUs) and it's not clear than any functional information or annotation was used. I would have liked to see more extensive use of the metagenome data, both for strain quantification and for functional interpretation, as is briefly alluded to in lines 360-367.

The process used to score and select combinations needs to be much better explained. Ideally, software used for this would be shared with the community to allow others to develop their own media modifications based on the data.

Some specific comments:

Line 83: "To enhance the recovery of fastidious gut microbes, we modified the medium's composition by introducing hemin, vitamin K1, and antioxidants." Why would these enhance the recovery of fastidious gut microbes? Has this been shown before or was it just hypothesized?

Line 103: "The fifth group contains seven host-derived metabolites known to modulate the gut microbiota and host physiology: primary and secondary bile acids." I found this phrasing a bit confusing – are all seven modifications bile acids? If so, perhaps "The fifth group contains seven primary and secondary bile acids, host-derived metabolites known to modulate the gut microbiota and host physiology" would be correct?

Figure 1: This figure leaves out the combinations entirely, which is especially confusing when Figure 2 includes results of

the combination experiments before those are described in the text. At the same time, nothing is said in the legend about the parts of the figure labeled “Benchling” and “Databases” which are hardly mentioned in the text. Could the figure be modified to include some indication of where the combinations fit in, or the title modified to note that it only covers the “benchmark” experiments?

Figure 1: The legend mentions 612 plates but doesn't indicate how this number was reached. Based on my reading of the method it appears to be from 6 donors X 51 cultivation conditions (50 plus base) X 2 dilutions. I think this is worth spelling out, particularly the two dilutions and the lack of any replicates.

Line 131: “Notably, Firmicutes, mainly Clostridiales species incertae sedis, were only detected in our stool samples, possibly due to metagenome assembled genomes (MAGs).” I couldn't understand what this was intended to say, especially since MAG assembly wasn't part of the scope of the work in the manuscript. I assume the failure of these species to appear in the cultivations simply means they aren't amenable to the cultivation conditions used.

Line 136: “...our cultivation approach can support growth of diverse taxa that is inaccessible to cultivation-independent methods.” Since whole metagenome sequencing was used to characterize both the stool communities and the cultivation products, clearly these organisms are accessible to “cultivation-independent methods” like metagenomics, but are presumably too low in abundance in the original stool samples to be detected at the depth of sequencing used.

Figure 3: Are the values displayed here treating each dilution as a separate data point, or combining the results of the two dilutions for each donor / media combination?

Line 159: “...PD variation was also noted among media modifications, particularly in histidine, chloramphenicol, vancomycin, CA, GCA, caffeine, ethanol, fluoride, nanoparticles, 10 dilution, and pH4, which were consistently beneficial across different groups.” I couldn't see anything distinguishing about this list of modifications in Figure 3B – many, but not all, of them have 95% confidence intervals that cross over Faith PD index of 10, but their average PDs are not necessarily higher than some of the other modifications not on the list, like tryptophan, mucin, or isobutyrate. And does “beneficial” mean “increasing diversity” in this case? What exactly is the comparison to?

Line 164: “Relative abundance was dominated by Bacteroidetes or Proteobacteria across most modifications (Figure 3C).” I couldn't discern any information about phylum-level relative abundance from Figure 3C, and in fact a much larger number of the families listed in that figure are Firmicutes than Proteobacteria. If phylum-level abundance is going to be referenced, perhaps this needs to be included in a supplemental figure.

Line 168: “...suggesting a link between media modification, microbial community composition, and PD increase.” This seems like kind of an obvious statement, that media composition will affect community diversity and composition; was any deeper insight obtained?

Line 170: This paragraph makes a lot of assertions about particular modifications leading to more or less phylogenetic diversity, referencing Figure 3C. Is this specifically family-level diversity (as compared to the PD index in 3B), and is there any statistical significance associated with these statements? Many of the statements here seemed anecdotal and given the lack of replication may not be reproducible.

Line 180: “Lachnospiraceae, Oscillospiraceae, and Ruminococcaceae species, known for health-promoting capabilities, were selectively enriched in multiple conditions...” I assume this is referencing Figure 3C, but what I understand Figure 3C displays is total family-level abundance of these groups. Did individual species within these families also consistently increase with these modifications, which isn't quite the same thing since the family-level increase could be due to a smaller set of species? Also it merits mentioning that these three families displayed similar abundance profiles based on the clustering shown in Figure 3C, although the distance metric and clustering method used for that figure are not clearly stated as far as I can tell.

Line 183: “Similarly, ethanol treatment, caffeine, nanoparticles and fluoride are also enriched for these taxa.” Does “taxa” refer to just Lachnospiraceae or all three families mentioned earlier in the paragraph? And how does this add to the first sentence, which already mentions caffeine, ethanol, and fluoride as enriching for the three families?

Line 187: Here, as in line 168, “suggesting” doesn't seem like the right phrasing as donor and media are clearly expected to influence taxonomic composition.

Line 199: “A principal component analysis (PCA) revealed 40.8% variance could be explained by the first two axes (Figure 4A), highlighting that the baseline composition of donors had a strong influence over the distance between samples.” For me this doesn't follow – the axes have no specific relationship to the donors so anything could be driving the differences between communities based on this display. I think some other statistical test is needed to make this point, like the permanova test mentioned later in the paragraph.

Line 201: “a Mantel test was used, showing significant correlations between the donor distance matrices (Figure 4B).” This sentence was meaningless to me. What is that test assessing and what is a “donor distance matrix”? Is this just showing that there are trends in relative abundance that are consistent across gut samples? Is Figure 4B based on stool profiles only, not cultivated communities?

Line 206: “Nonetheless, some modifications also exhibited a significant effect on composition at genus level ..., with a subset of samples demonstrating taxonomic diversity beyond the donor baseline (Figure 4A, marked in lavender).” How are the lavender circles in 4A calculated and which points (if any) correspond to the “donor baseline” (which I assume is the stool sample profile)? They appear to be arbitrarily drawn at the edges of the plot, and without the plots being annotated with modifications the next sentence, “Interestingly, most of those samples are associated with a more diverse taxonomic enrichment compared to base medium (Figure 3B and C),” cannot be assessed.

Line 213: What does “superior” mean in the context of abundance variation – greater or less?

Line 214: “...although donor influence was in general the major contributor to final taxonomic composition observed, certain modifications exerted a notable impact at the genus level.” Which data demonstrate this and what is the statistical significance, particularly the next statement that some modifications had “a more substantial influence than the donor effect”?

Lines 237-238: Bacteroides and Proteobacteria are twice referred to as “genera” but Proteobacteria is a phylum. Based on Figure 5B I think the reference is actually to an unnamed genus within the Proteobacteria but that needs to be clarified. And I couldn't see any data supporting the statement that “these two genera are usually present and dominant in most modifications, while other genera vary more broadly in occurrence” since no genus-level abundance data are presented.

Similarly, the next sentence, “Interestingly, we observed a clustering of Firmicutes and Actinobacteria, revealing potential similar media preferences,” lacks a figure reference and doesn’t appear to be supported by Figure 5B, where genera within the Firmicutes, Bacteroidetes, Proteobacteria and other phyla are intermingled in the tree with the relatively few Actinobacteria genera. I think all of these statements need better support if they are included, but they also need some biological interpretation to be useful and the current Discussion lacks any discussion of phyla or genera and their media preferences.

Figure 5A: It would help to have some explanation of how this plot was generated. Is this just a standard boxplot, but with individual data points plotted on the right hand side?

Line 258: “We calculated and ranked combinations based on mean relative abundance and absolute count for targets across modifications for each donor and each combination.” A better explanation is needed for how combinations were scored and ranked. As far as I can tell the scoring and ranking process is not described in the methods. An attempt is made in Figure 6A but it isn’t understandable and definitely wouldn’t be reproducible.

Line 268: It would be helpful to clearly state that *Collinsella* relative abundance is the metric being used here for improvement.

Line 275: “A combination of 4 media modifications (acetate, pectin, 10X dilution, pH6) was able to achieve 10 to 20-fold enrichment in dopamine and DOPAC pathways over the mean relative abundance in the base medium.” This implies that these pathways were directly assessed in the metagenome data – if so how? The Figure 5C legend says “Bar plot for the fold change enrichment over Base for strains associated with dopamine and DOPAC synthetic pathways,” which implies that mOTUs were used as proxies for these activities. Again, no details are provided in the methods on how these pathways were assessed and how combinations of modifications were predicted and tested. This needs to be much more clearly explained.

Figure 6D is very hard to follow.

Figure 6E is so briefly explained it’s hard to grasp its purpose – how was this medium designed and what was the intended outcome? The methods do not include an explanation.

Version 1:

Reviewer comments:

Reviewer #2

(Remarks to the Author)

I thank the authors for the time and effort they took to reply to my comments and revise their manuscript. This is much appreciated. The following comments are intended to clarify some of my earlier points and to offer a few more suggestions, which I hope the authors will find useful.

(1) Figure 3B, data missing for rightmost two categories “total on media & stool” and “total on media only”: The authors reply that it would not be meaningful to calculate PD for these two categories. Still, I do not understand. PD is a measure that is calculated on each sample, in the same way mOTU richness is calculated on each sample in Figure 3A. In panel 3A, the points are shaded by donor, while in 3B the points are shaded by modification type. The authors have made a color for “stool sequencing”, why can’t they make a color for the last two categories? Relatedly, the Results text concerning Figure 3B refers to something called “PD variation”; for example, “[o]verall, the highest PD variation stemmed from inter-donor initial composition.” But the variation in PD among donor stool samples – presumably depicted by the size of the rightmost boxplot – is certainly not the largest of those shown. The figure caption says PD is calculated “across modifications for all donors”, which doesn’t really explain what each point is. And PD is not mentioned at all in the Methods. Altogether, I remain worried there’s been some confusion, miscommunication, or mislabeling with respect to Figure 3B.

(2) Figure 4A, hand-drawn ovals: I understand better now the purpose of the hand-drawn ovals, but I think the figure caption should further specify that an arbitrary number of samples appear within each hand-drawn oval. (In fact, a few samples are half in, half out.) The legend should also probably note that hand-drawn ovals don’t highlight any particular type of modification.

(3) Figure 4B, Mantel tests: I appreciate the authors confirming I guessed correctly about the inner workings of their Mantel test. But I don’t think the results show that conditions tend to enrich for the same genus in a particular medium; a distance is a distance, not a direction. Donors 1 and 2 might each be pushed far by modification A, but it could be that modification A selects for completely different genera in donor 1 (versus donor 1 base) than in donor 2 (versus donor 2 base).

It seems like what you’d like to visualize within each modification is (1) the average of the pairwise distances between different donors, versus (2) the average of the pairwise distances between each donor and their own base. [It just seems like you’re trying to identify modifications in which donor composition moves far (high #2) AND in the “same (compositional) direction” (low #1).] I suspect this is what Figure 4C is sort-of getting at, but the writing and labeling around 4C remain difficult for me to parse.

(4) Figure 5A, samples or taxa: When I first viewed this plot, I assumed that each point represented a Hellinger distance between a pair of samples, and that taxa were glommed at the given taxonomic level prior to calculation of the sample-to-sample distances. But now, noticing that Phylum has many fewer points (and that my prior interpretation wouldn’t make much sense given the question posed by the authors), I realize that maybe points represent Hellinger distances between pairs of taxa. In either case, there’s an awful lot of (non-independent) observations here. Perhaps the authors could display

for each taxon(?) the average of its distances to all others? In general, I would encourage the authors to explain in greater detail the entities and comparisons displayed.

(5) Figure 6CD, relative abundance versus fold change: Okay, it just seemed odd to switch from one (panel C; rel. abund.) to the other (panel D; fold change). By the way, it would be helpful to edit the panel D title to something like “Dopamine & DOPAC biosynthetic gene cluster”, or similar.

(6) Line 311, Lactobacilli and Bifidobacteria: Here and throughout, the authors are encouraged to review the formatting of informal names versus genus names, etc. For example, here the genus names are *Lactobacillus* and *Bifidobacterium* (italicized), while the informal names are lactobacilli and bifidobacteria (non-italicized).

(7) Line 471, “without obtaining a loan”: I believe the authors intend the word “lawn” here, rather than “loan”.

(8) Availability of data and code: Thank you for providing a reviewer link to your sequence data submission. The authors are further encouraged to make their analysis code available as well.

Reviewer #3

(Remarks to the Author)

I have gone through the authors' responses and the revised manuscript and many of my concerns have been addressed, but several have not and need further work. Below are my remaining points that haven't been fully addressed:

Line 84 – In my previous review I asked why hemin, vitamin K1, and antioxidants were expected to enhance the recovery of gut microbes, and the authors responded with multiple citations. Yet the manuscript itself remains unchanged and these references have not been added – I assume other readers might have the same question and I don't see why the authors wouldn't want to clarify this point.

Figure 1 legend – In response to my question about the number of plates examined, the legend was modified to read, “After seven days, the 612 (6 donors x 51 conditions x 2 dilutions) plates were scraped to recover...” Yet elsewhere in the manuscript it is noted, “Only one dilution plate out of the two created for each sample was scraped, leaving a single dilution per sample to prepare for sequencing.” Doesn't this mean only 306 plates were actually scraped?

Lines 166-173: the authors have added Supplementary Figure 1, addressing my concern that the statement about Bacteroidetes or Proteobacteria dominating most modifications wasn't supported by the data presentation. However, the sentence still reads “Relative abundance was dominated by Bacteroidetes or Proteobacteria across most modifications with exceptions in ... where a reduction in relative abundance for these families can be observed.” Part of my concern was that this sentence appears to refer to the phyla Bacteroidetes and Proteobacteria as families, but I think the authors intend to say that families within these phyla decrease in relative abundance so the sentence needs to be rephrased.

Lines 174-184: In response to my question about statistical significance, the authors acknowledged, “No statistical tests were conducted on these observations. The observations were intended to be descriptive, highlighting patterns that may warrant further investigation.” Yet the phrasing of this paragraph remains unchanged and is still phrased as though these are well supported (e.g. “histidine and tryptophan enriched for more diverse taxa than tyrosine” and “taxa belonging to Eggerthellaceae, Clostridiaceae, Bifidobacteriaceae were positively enriched in CA, GCA and TCA cultures”). I would think that the six donors would provide sufficient sample numbers for statistical tests (for example, were those families consistently enriched in those treatments for all donors) but if these observations are not statistically supported they should either be removed or clearly noted as merely observational.

Line 185: “Lachnospiraceae, Oscillospiraceae, and Ruminococcaceae species ... were selectively enriched in multiple conditions...” In response to my query about this statement relative to the data presented in figure 3C, the authors acknowledged that only family-level abundances were presented and shown to have these patterns, yet they failed to remove this reference to species that implies that individual species showed these patterns.

The authors also said in their response, “Regarding the clustering in Figure 3C, we acknowledge that the distance metric and clustering method were not explicitly stated, and we will clarify these details to enhance transparency.” As far as I can tell, this was not done.

Line 191: In my previous review I noted that “suggesting” was not the best term for something that seemed very clear, and the authors responded “We have rephrased it using the term ‘indicating,’ which we believe is more appropriate.” This also appears not to have been done since this line still reads “...suggesting media modification and donor-dependent effects on taxonomic composition.”

Line 203: I and another reviewer both noted that the percent of variance explained by the PCA axes provides no insight into the factors driving the differences and the authors agreed. But this line still reads, “A principal component analysis (PCA) revealed 40.8% variance could be explained by the first two axes, highlighting that the initial composition of donors had a strong influence...” I think simply changing the word “highlighting” to “and highlighted that...” would clarify that this is apparent from the PCA itself, not from the percent variation explained by the axes.

Line 214 and Figure 4C: “...the average ratio of within-donor variance to between-modification variance (Figure 4C and Methods) was below 0.1.” The y axis on Figure 4C is labeled “Variance between donors over variance between genera” which to me is meaningless; this is also in the legend (“Ratio of the variance within donors to the variance between genera for each modification”) but I think “genera” should be “modification” in both these places.

Line 259: “...we observed a clustering of genus belonging to Firmicutes and Actinobacteria, revealing potential similar media preferences.” I noted in my previous review that no such clustering was apparent in Figure 5B and the authors responded “We believe that the statement regarding the proximity of Firmicutes and Actinobacteria in the clustering is supported...” While species of these phyla do both occur in the same very large cluster, they are spread out and intermingled with members of other phyla which I don't think is well described by “clustering” – maybe a rephrasing like “... ”

we observed that genera belonging to Firmicutes and Actinobacteria were all part of the same large cluster, revealing..." would be more clear.

Figure 5A: The authors responded to my query by confirming the plot is "a standard boxplot with individual data points displayed on the right side," but did not clarify this in the manuscript itself.

Lines 499-512: I continue to find the description of the ranking and selection process inadequate. Maybe this descriptive text could be supplemented by a formula indicating the actual calculation? And perhaps one of the examples, for example the *Colinsella* story, could be fully explained in the supplementary material with real numbers?

Lines 291-298 and Figure 6D: In my previous review I asked whether abundance of dopamine and dopac pathways was assessed directly from metagenome data or inferred based on species abundance, and the methods were clarified to make it clear that this analysis was based on functional gene abundance in the metagenomes. Yet the legend for Figure 6D still says "Bar plot for the fold change enrichment over Base for strains associated with dopamine and DOPAC synthetic pathways..." This should say "genes," not "strains."

One other minor comment - Line 125: "...were obtained for both cultured communities, as well as the original stool sample." "Both" is redundant here.

Version 2:

Reviewer comments:

Reviewer #2

(Remarks to the Author)

My concerns have largely been addressed. Thank you!

There remains one point of disagreement concerning Figure 3. In my original review, I asked why values for the categories "Total on media and stool" and "Total on media only" were present in Figure 3A (OTU richness) but absent from Figure 3B (Faith's PD). I asked this because the plotted entities are the same across Figures 3A and 3B (each point is a Donor) and because the measures – OTU richness and Faith's PD – can each be calculated on any sub-set of taxa.

The authors initially replied that for "Total on media and stool" and "Total on media only", there was "no value" for Figure 3B because Faith's PD would "not be meaningful to calculate" for these two categories. I did not fully understand this reply, because the only requirement for calculating Faith's PD on a sub-set of taxa is that all taxa appear together in a phylogenetic tree. The authors display this tree in Figure 2. Thus, upon re-review, I asked for further clarification.

The authors have now replied that Faith's PD must be calculated on a "single community or sample", that calculating Faith's PD is "not appropriate" in these cases because taxa are aggregated across samples, and that Faith's PD calculation "assumes that taxa are part of a single, cohesive ecological community where phylogenetic relationships are meaningful in the context of the community", and that doing so would "violate this assumption, making such PD calculations statistically invalid and their interpretation unreliable". And so on – e.g., Faith's PD "require[es] taxa to be part of a coherent ecological unit for phylogenetic relationships to be meaningful".

These statements are false. Phylogenetic relationships are not context- (or community-) dependent and Faith's PD makes no such assumption. In fact, the paper in which Daniel Faith introduces the metric (Faith 1992) is concerned with conservation priorities, reserve design, and assessing the phylogenetic diversity of sub-sets of threatened species – not community ecology. I refer the authors to this paper below. Faith's PD is the sum of branch length leading to a set of taxa in a phylogenetic tree. It is not an estimator; not an index.

What is true – OTU richness and Faith's PD may not agree; they may not yield the same pattern or conclusion. Two sets of taxa may have identical OTU richness (say, 5 taxa each) but very different Faith's PD because one set has five very closely related taxa while the other has five very distantly related ones. Thus, comparing the two metrics may yield interesting insights into the nature of diversity.

If the authors do not think it reasonable or meaningful to assess alpha diversity for sets of taxa representing more than one sample or community, that's fine. (But are they even doing that? The community exists at the level of the Donor.) It just doesn't make sense to show one metric and not the other. It would seem reasonable to me to show both metrics or neither metric for the categories "Total on media and stool" and "Total on media only".

Faith, Daniel P. "Conservation Evaluation and Phylogenetic Diversity." *Biological Conservation* 61, no. 1 (1992): 1–10. [https://doi.org/10.1016/0006-3207\(92\)91201-3](https://doi.org/10.1016/0006-3207(92)91201-3).

Reviewer #3

(Remarks to the Author)

I believe my remaining concerns have largely been addressed. I am still concerned that the observations in lines 178-188 are statistically unsupported; while the wording has been toned down I think it would still benefit the reader to include p-values or note that additional experiments would be needed to validate the trends observed. For example, the paragraph could start, "In the aromatic amino acids group, some qualitative observations that merit further exploration are that histidine

and tryptophan..."

I have not gone through the manuscript sentence by sentence to make sure all assertions are statistically supported, but I encourage the authors to make sure this is the case.

Reviewer #1 (Remarks to the Author):

Thank you for giving me the opportunity to review this article. Over the past 12 years, culturomics has become a key technique for studying the digestive microbiota, thanks in particular to the prospects it offers for modulating the microbiota. The authors propose a very interesting work. I find this manuscript particularly well written and clear. I have several comments:

Comments:

1.1 I suggest that the authors describe their supplementary material culture protocols even more exhaustively, so that the scientific community can reuse these techniques.

This comment is in line with the feedback from the other reviewers, and we have expanded the methods section significantly to accommodate this request. (Materials and methods section, line 400-540)

1.2 Readers may be disappointed by the fact that culture guides metagenomics but strains are not obtained. I think the authors need to discuss this limitation more precisely and could present it as a preliminary study. Indeed, only pure culture can enable a therapeutic approach.

We thank the reviewer for this feedback. We acknowledge the fact that individual strains were not isolated in this study or evaluated for potential therapeutic applications. However, the primary objective of our study was to evaluate cultivation conditions that could enrich target taxa, addressing the current bias in most media towards Bacteroides species and Enterobacteriaceae, which do not fully represent the diversity of the gut microbiome. This manuscript demonstrates that adjusting a few media ingredient in a commercially available media (mGAM) can either increase or inhibit particular taxa. This approach is generally applicable, and we believe it would be of interest to the research community.

This study lays the foundation for future microbiome research, including isolation projects and the evaluation of strains for therapeutic use. Finally, we agree a limitation of our study is the relatively small number of donors (n=6), a trade-off decision made between the number of conditions tested and the cost of sequencing depth. Future studies could benefit from including a larger number of donors and testing a reduced number of conditions to test more specific hypothesis.

1.3 A recent article (Nat Biotechnol. 2023 Oct;41(10):1424-1433. doi: 10.1038/s41587-023-01674-2. Epub 2023 Feb 20.) discusses the automation of culturomics. How do the authors position their work in relation to these perspectives?

Thank you for highlighting the recent article by Huang et al. (2023). We have reviewed their work and noted their emphasis on automation in culturomics. However, it is important to point out that the approach by Huang et al. does not include an enrichment step, which can result in extensive colony picking to identify less abundant strains. In some cases, this may even hinder the recovery of certain strains, as observed in our study where some strains did not grow in common commercial media like mGAM, only in some of the modifications. The discussion point about automatization in culturomics has been added to the discussion section (Line 388-390):

“Furthermore, the methodology presented in our manuscript, when combined with automation, could significantly enhance the recovery and isolation of less abundant taxa, advancing culturomics studies and help building large culture collections.”

Reviewer #2 (Remarks to the Author):**Summary**

The authors seek to develop generalizable approaches to improving the cultivation of gut microbes. They apply a panel of enrichment conditions (modifications), individually and in selected combinations, to stool samples collected from six healthy adult volunteers. Stools and enrichments were surveyed using metagenomic sequencing. Compositional variation among enrichments (scraped plates) was explored with respect to donor (inoculum) community composition; the type(s) of modification(s) made to the base medium; and bacterial taxonomy. The strengths of the work include the systematic exploration of a wide range of individual modifications while holding all else constant, and the inclusion and consideration of different donor (inoculum) communities. Several areas where the work could be improved would be (1) a clearer, more specific, more detailed statement of the problem or question addressed by the research, and the bigger-picture goal; (2) clearer and more thoughtful communication of results and rationale for selecting or singling out individual modifications, taxa, and functions for focused study; and (3) supplying additional methodological details. The following comments are intended to help the authors improve the communication of their work.

We have significantly revised the main text to improve clarity. This includes a more detailed and specific statement of the problem and clearer explanations of our rationale for focusing on modifications, taxa, and functions. Concrete examples of these improvements are provided in the reviewer's comments below.

In addition, we have substantially expanded the relevant sections in the Materials and Methods, as well as the figure legends, to include the additional methodological details requested. We appreciate the reviewer's feedback and believe these revisions enhance the clarity and quality of our manuscript.

Comments:

2.1 Title: I don't think metagenomics "guided" the culturomics; rather, it was metagenomics-based? Metagenomics was the read-out. Also, I don't feel the methods were really that "targeted", at least not from an a priori design perspective that was communicated.

Thank you for this remark. We considered the appropriate title for this manuscript to be "metagenomic-guided" because the metagenomic sequencing data generated in the experiments was used to guide the design of media combinations for the enrichment of specific taxa and functions.

2.2 Methods:

- 2.2.1 Line 424-425: Two serial dilutions were used to inoculate each medium. What became of the two dilutions? There's no other mention of them.

Thank you for bringing this to our attention, we have added a descriptive statement related to the dilutions in the materials and methods (Line 459-463):

“2 serial dilutions (100X and 1000X) per stool sample were selected to inoculate (100uL) the chosen media prepared earlier and spread with sterile glass beads. Chosen dilutions depend on the initial cryostock cell density and preservation. Dilutions need to be tested on base medium prior to the cultivation campaign (mGAM without modification) to select the most suitable dilution”

- 2.2.2 Line 429-434: Understanding the work depends a lot on understanding these Methods. Please address the following:
 - 2.2.2.1 The presence of confluent colonies resulted in the elimination of the whole plate, or simply avoidance of the confluent colonies while scraping other regions of the plate? And what’s the problem with confluent colonies, anyway, given that isolation was (seemingly) not a goal of this study?

Multiple reasons were behind this choice. First, we made the assumption that selecting plates with the maximum of biomass while not obtaining a loan might enable the growth of underrepresented taxa that could be overwhelmed on a plate with a loan. Second, while isolation was not the primary objective of our study, a direct continuation of this work would be to leverage this technology for isolation. Therefore, it seemed appropriate to be using a protocol close to that potential experimental goal. Finally, one of the original project’s goals was to collect images of plates to build a classification model based on the morphology of individual colonies. However, due to technical issues, this aspect of the study was not pursued further.

2.2.2.2 Was the entirety of each plate scraped into a single 2 mL tube, along with the 2 mL of PBS used to facilitate colony detachment? Or was each colony of each plate placed into a separate tube? This is relevant to understanding the meaning of “relative abundance” in the context of the plate scrapes.

The entirety of one of the two plated dilutions for each sample was scraped into a single 2 mL tube with PBS. This approach was used to collect the samples from each plate as a whole. We have revised the main text to clarify this (line 467 to 471):

“7 days after inoculation, all the plates were imaged and cultures were manually scraped on the entire plate surface to collect all colonies into 2mL Eppendorf tubes, using 2 mL PBS to facilitate the colony detachment. Only one dilution plate out of the two created for each sample was scraped, leaving a single dilution per sample to prepare for sequencing. The scraped plate was selected to achieve the maximum cell density that can be recovered without obtaining a loan”

2.2.2.3 What was the purpose of imaging if all biomass from each plate was scraped into a single tube? Or was each colony tracked?

We thank the reviewer for this observation. The initial idea was to build a classification model and link the morphology (from images) with genetic information (from the metagenomic sequences) to see if any useful information could be deduced from the imaging, as well as the ability to track the colonies for implementation in an automated setup. Due to technical challenges this was not

implemented in the final manuscript, and we have now removed this part from figure 1.

- 2.2.3 Line 459: Why was the Hellinger distance selected here? Were the results robust to choice of distance metric? Also, what Methods (packages) were used for PCA (PCoA?); PERMDISP & PERMANOVA.

We chose the Hellinger distance because it offers a balance between linearity and resolution, expresses data as relative abundances per site to mitigate sampling discrepancies, reduces high abundance values asymmetrically through square-root transformation, and conforms to Euclidean space¹, facilitating the use of distance-based calculations and dimensionality reduction tools such as PCA, UMAP, and clustering algorithms. We added a paragraph explaining these choices in the methods section.

- 2.2.4 Line 466: The BioProject accession does not exist in a public-facing format. If the data are pending release, then a reviewer link should be made available. The authors are also encouraged to plan to make their analysis code available.

We have fixed the access and made the BioProject accessible using the follow link: <https://dataview.ncbi.nlm.nih.gov/object/PRJNA1077691?reviewer=kkk8qgid6jue1me3j1c4ddijje>

This has also been updated in the main text line 545. "All sequencing data was deposited in SRA database (Bioproject accession number: PRJNA1077691, <https://dataview.ncbi.nlm.nih.gov/object/PRJNA1077691?reviewer=kkk8qgid6jue1me3j1c4ddijje>)."

2.3 Results:

- 2.3.1 Throughout: The following terms are used: stool, scrapes, plates; base, baseline, benchmark; modifications (unique or individual); and combinations. It gets confusing. Please define your terms and use them consistently.

We thank the reviewer for pointing this out and agree that some terms are incorrectly used interchangeably. We have updated the manuscript to streamline this, and rephrased "benchmark" to "single modification cultivations" to avoid confusion and better connect with the nomenclature used to describe "combination modifications".

While many of the other terms used are commonly understood within the field, we have carefully reviewed and streamlined their usage to improve overall consistency throughout the manuscript. We hope these changes enhance the clarity of the manuscript.

- 2.3.2 Line 131: If a given mOTU was detected in the original stool of 6 out of 6 study participants, but was detected in only one participant's collection of benchmark cultivations, how many times was it considered absent?

Recovery is assessed in absolute terms across the entire sequencing dataset, meaning that raw sequencing data and cultivation data are pooled and compared together. If a sample is detected during cultivation but not in the corresponding donor's raw sequencing data, it is still considered "recovered." Our goal was not to demonstrate the

recovery of a specific percentage per donor, but rather to illustrate that a certain proportion of overall diversity can be recovered. Additionally, some diversity may be detected in cultivation that is not present in the raw sequencing data.

- 2.3.3 Line 137 “that is inaccessible”: Your result is that cultivation is inaccessible to cultivation-independent methods? This seems obvious. Maybe you mean that the diversity was not accessed, as in:

- “Our cultivation approach enriched for diverse taxa that were undetected at the depth of sequencing (arbitrarily) used in our cultivation-independent surveys.”

We agree with the reviewer that this is a better explanation and have implemented this change (Line 139-141):

“These findings, consistent with culturomics studies^{17,28}, indicated that our cultivation approach can support growth of diverse taxa that were undetected at the depth of sequencing used in our cultivation-independent surveys”

- 2.3.4 Line 139: On the one hand, it’s perhaps not surprising that different culture conditions enrich for different taxa. On the other, it’s certainly interesting to interrogate how single modifications might shape or shift the enrichment landscape. I wonder if additional visualization might help develop intuition about this selective landscape. For example, approaches like RDA or CCA which could reveal modifications (or groups of modifications) that most strongly associate with enrichment or depletion of particular taxa (or groups of taxa) across multiple participants.

We agree with the reviewer on this point and have previously assessed multiple methods for visualizations and found it difficult to choose which way to present the data. Many confounding factors like donor or partial data (sequencing depth) make it hard to get a clear picture with the suggested visualisations. Nevertheless, it is essentially what we try to highlight in the rest of the paper, which modifications (or groups of modifications) most strongly associate with enrichment or depletion of taxa. This is a vast subject specific to each target to properly communicate. The raw data has will be made available together with the publication of this manuscript so that others can analyse the data with their particular target of interest in mind.

- 2.3.5 Line 149: Start of sentence (or more?) is missing. Feels like maybe a whole paragraph has been accidentally deleted.

We do not believe this is the case, but we have rephrased the beginning of the paragraph to improve readability. Perhaps the reviewer was referring to the sentence on line 196-197, which we have also removed now as this was indeed a formatting error.

- 2.3.6 Figure 3B: Boxplots missing for last two categories at right? Also, the color difference between amino acids and antibiotics is undistinguishable to me, here and throughout.

There is no value for the last 2 because they would represent the PD index for the total for stool+media and media only, which is not meaningful to calculate here. We agree that these colours were too close to each other, and have updated the colour coding throughout the manuscript to improve this.

2.3.7 Figure 3C: Would love to see an additional panel showing participant- and species-resolved data for the three modification-family pairs with big increase (the yellow cells in panel C).

We thank the reviewer for their suggestion and agree that this could be an interesting analysis. However, we believe that the manuscript already presents a substantial amount of data, and adding further information might dilute the core messages we intend to communicate. We will make the relevant data publicly available, allowing the scientific community to explore and analyse it further, according to their specific interests.

- 2.3.8 Line 164: Not sure this statement follows from Figure 3C. At least it's not obvious to me.

When looking at Figure 3b and 3c, it can be observed that PD values (3b) and number of enriched taxa (green squares in 3c) are higher for histidine and tryptophan than for tyrosine (11/14 vs 8 squares). We have updated the text accordingly to make it clearer (line 174 -177)

"In the aromatic amino acids group, histidine and tryptophan enriched for more diverse taxa than tyrosine, with tryptophan selectively enriching for Clostridiaceae, Bifidobacteriaceae and Akkermansiaceae species compared to the base medium (PD values in figure 3B and number of green squares in 3C)."

- 2.3.9 Line 180-181: Would it be more accurate to say they've been associated with states of health?

Thank you for pointing this out. We agree that the phrasing can be made more accurate. The term has now been changed throughout the manuscript to better reflect an association with health status, rather than implying health promotion.

- 2.3.10 Line 185 "significant": Please state the details of any statistical tests.

The "significant" description was in this case used in the non-scientific meaning to describe a large amount of variation, but we omitted the word now as we agree this is incorrectly used. We thank the reviewer for bringing this to our attention.

- 2.3.11 Line 199-200: The percentage of variation explained says little about the source of the variation. If you observe clustering by participant, meaning that media modifications don't completely override that signal, then one might say that donor composition is reasonably stable to most selective regimes. (In some ways, it must be: there's no alternative source of microbes – none other than the unsampled tail of the rank-abundance curve.)

We agree that the percentage of variation explained by our data provides limited information about the specific sources of variation. Indeed, if media modifications do not completely eliminate clustering by donor, it suggests that donor composition remains relatively stable across different selective conditions. This stability implies that the variations we observe are predominantly due to media modifications as we discuss in the manuscript.

- 2.3.12 Line 200-201 and Figure 4A: What is called “baseline” – the stool or the base media? Also, unclear whether “baseline” samples even appear in Figure 4A (given the caption). And finally, because the (Figure 4A) plot regions representing the highest values of PC1 and PC2 contain samples (ostensibly from certain modifications) from numerous participants, it seems unlikely that baseline variation among donors has the strongest influence on the compositional variation depicted on this plot. I think what you’re trying to depict is that most modifications reflect baseline clustering of samples by participant (in other words, the modifications cluster near the stool or base); while just a few modifications (the extremes noted here) reflect clustering by modification.

We agree with the reviewer that this was not properly addressed in the figure or the main text, and understand the confusion between stool and what was referred to as “baseline”. We have updated the figure to contain a black highlight around the points that correspond to the baseline medium for each donor and corrected the text and figure legend to avoid any ambiguity. We hope the reviewer agrees with this revision.

- Would it be possible to highlight each participant’s unaltered stool sample with a different shape or outline? Or perhaps provide a supplement in which the samples are colored by the few modifications you wish to highlight? (Or label the ovals so the reader might understand which modifications are highlighted?)

We agree this comment and have changed the figure to have a brown circle around the base for each donor, highlighting where the points should cluster

- 2.3.13 Line 201-202 and Figure 4B: I understand what a Mantel test is, but I don’t understand what you’ve done here. Is this showing the degree to which pairs of participants had similar responses to the cultivation regimes, in terms of which conditions pushed the composition more or less away from their own baseline? For example, participant 2’s within-participant distances are most highly correlated with participant 3’s within-participant distances?

The reviewer has correctly understood this analysis. We have updated the figure legend to clarify how the analysis was performed. In line 232 – 238

“B. Heatmap of the Mantel test results between donor distance matrices using the Pearson R correlation coefficient (p -value <0.01 for all correlation coefficients). The Hellinger distance between modifications at genus level is calculated for each donor, these distance matrices are leveraged in a mantel test to calculate correlations between donors. A positive correlation means that there is a correlation between distance matrices for a set of donors, and that conditions tend to enrich for the same genus in a particular medium.”

- 2.3.14 Line 207: To which comparison does this p-value apply? Is this a PERMANOVA of “stool” (or “base”) versus each “mod”, or other? Please take greater care to explain your results.

We have updated the entire paragraph to be more concise and expand on the statistical parameters used in each comparison (Line 206- 222):

“To further characterize the relationship between donors, a Mantel test was used,

showing significant correlations between the donor distance matrices (Figure 4B). Finally, PERMANOVA analysis confirmed the significant impact of donor composition on genus-level microbiome composition ($F=20.55$, $p<0.001$) without affecting sample dispersion (PERMDISP: $F=1.43$, $p=0.204$, 999 permutations). These results indicate that the baseline donor composition substantially influences the microbiome composition across most medium modifications. However, modifications also demonstrated a significant effect on genus-level composition (PERMANOVA: $F=2.42$, $p<0.001$) without affecting sample dispersion (PERMDISP: $F=1.01$, $p=0.189$, 999 permutations), with some samples exhibiting taxonomic diversity beyond their initial donor composition (Figure 4A, marked in lavender). Overall, the average ratio of within-donor variance to between-modification variance (Figure 4C and Methods) was below 0.1. This indicates that the genus relative abundance variation due to modifications was, on average, ten times greater than the variation attributed to differences between donors. Some modifications, such as caffeine, resulted in different taxonomic diversity compared to the base medium (Figure 3B/C) but showed a stronger influence from the initial donor composition (higher ratio value in Figure 4C). In contrast, other modifications, such as CDCA, had a more pronounced influence on taxonomy relative to donor influence (lower ratio) and exhibited lower diversity.”

- 2.3.15 Line 209: I do not see how the statement on this line is supported by Figures 3B and C, because you have not mentioned which modifications or samples these are (at the periphery of Figure 4A).

We have updated the whole paragraph to make the message clearer for the reader. This statement is not present anymore in the text (see quote above).

- 2.3.16 Line 211: Which lavender region (“cluster”)? There are three. Are these confidence ellipses? Or were they drawn by hand? Why mention (whichever one is) CDCA, and not the others?

The ellipsis was drawn to highlight regions where some sample cluster away from most of their donor location, e.g. many samples for donor 2 (red) cluster bottom left, whereas a few conditions occur in the upper and right circle. This representation was chosen to give a sense of that phenomenon, without over cluttering since the specific condition position, more that it's detached from the donor cluster, ending up therefore in one of the lavender circles. We have updated the legend of the figure to specify that the lavender circles were drawn, not computed.

CDCA is mentioned not relative to 3A, but 3B/C as it is very selective independently from donors as seen in 3c

- 2.3.17 Figure 4C and lines 211-213: Figure 4C is not mentioned in the text. The sentence ending on line 213 might refer to Figure 4C, but the sentence is too confusing (to me) to reconcile with the figure.

Thank you for pointing this out. We apologize for the confusion. Figure 4C was indeed not referenced in the text; therefore, we have revised the text to improve clarity and correctly reference the appropriate figures. Lines 215-22:

“Overall, the average ratio of within-donor variance to between-modification variance (Figure 4C and Methods) was below 0.1. This indicates that the genus relative abundance variation due to modifications was, on average, ten times greater than the variation attributed to differences between donors. Some modifications, such as caffeine, resulted in different taxonomic diversity compared to the base medium (Figure 3B/C) but showed a stronger influence from the initial donor composition (higher ratio value in Figure 4C). In contrast, other modifications, such as CDCA, had a more pronounced influence on taxonomy relative to donor influence (lower ratio) and exhibited lower diversity.”

- 2.3.18 Line 231: It stands to reason that phylogenetic relatedness predicts ecological similarity.

Our results confirm that, in most of the conditions observed, phylogenetic relatedness does indeed predict ecological similarity. However, while phylogenetic relatedness can help predict ecological similarities, the reverse is not always true. Ecological similarity does not necessarily imply phylogenetic relatedness. For example, Bacteroides and Proteobacteria, which both thrive in the colon and grow well on mGAM, are phylogenetically distant despite their similar ecological niches. This illustrates that ecological similarity does not preclude significant phylogenetic divergence.

- 2.3.19 Line 236 (Figure 5B), highlighting a segregation: I don't really see it; not sure I understand.

In 5b, the top of the dendrogram shows a clear separation from all the rest (blue branch vs brown branch). We have updated the colour palette in all figures to increase the contrast and improve readability.

- 2.3.20 Line 257: taxon

Thank you. However, the term in this part of the text is meant to be used in the plural form as 'taxa.'

- 2.3.21 Line 266: *Collinsella aerofaciens* was present in five modifications ... all from the same participant?

We thank the reviewer for pointing out this part of the text was not clear. We have revised the text to make it clearer (Line 286 – 288):

“Since it was present in only five of our single modification cultivations (Figure 6B, green bars), we created a new medium that combined these modifications. This medium included pectin, inulin, 10X Base, lactate, and taurocholic acid.”

- 2.3.22 Line 269: I'm not sure I understand the goal of the approach, or the interpreted meaning of relative abundance in this enrichment context. Is the goal here to reduce the number of colonies one has to sift through before hitting this species (on a plate that enriches for many)? Is the goal to obtain an isolate from an individual patient?

If the initial condition supports growth, then reducing the number of colonies to pick becomes the target. However, if there is no observed growth on a traditional cultivation medium, finding the modifications enabling the target growth of even a low relative

abundance opens new ground to help recover and study inaccessible diversity. In the future, this methodology could support the study of a wider range of species in labs and potentially lead to new therapeutic strategies, including personalized treatments.

- 2.3.23 Line 276: Why expressed as fold-change whereas *Collinsella aerofaciens* data were expressed as relative abundance? Prefer the latter. Ditto for Figure 6E.

The aim is to give an easy representation of the abundance improvement using combination relative to base media, and the power of such approach. Using relative abundance makes it harder to grasp, as moving from say 0.01% RA to 1% represent a 100-fold improvement, which could enable the recovery when paired with a suitable pipeline, but still appear very low. Targets of interest for this kind of approach are generally undetected or extremely low RA, thus grasping what is the potential improvement compared to a conventional approach makes it clearer than a percentage value.

- 2.3.24 Lines 298-300: Yes, I think that's right. In other words, there's probably a lot more "non-detection" than true absence. I think this is generally recognized.

We agree with the reviewer that this is the general notion, but our study provides some data and arguments to this theory, as well as providing a solution and workflow for how to deal with this problem.

Reviewer #3 (Remarks to the Author):

This manuscript presents the results of a set of "culturomics" experiments in which microbes were cultivated from stool samples using a large number of variations on a standard medium, then characterized via shotgun metagenome sequencing to analyze the impacts on overall diversity as well as changes in the retrieval of individual species. The investigators found that even small changes in media composition resulted in changes in the diversity and composition of the organisms cultivated from multiple stool donors. They also found that combining modifications could result in further changes in cultivable communities, which were not always predictable based on the results of the single modifications. This is a nice study that could be of interest to a range of scientists, particularly those studying gut communities but also those studying a range of microbiomes. However, in many cases the results have only been minimally explored and the outcomes aren't always clear. A few overall points: The distinction between the "benchmark" experiments and the "combinations" is never clearly explained and has to be inferred from the descriptions, which is particularly hard when the combination results are included in Figure 2 but the combinations are not described in the narrative until line 256, well into the results section.

Thank you for pointing this out. We have updated the manuscript throughout to clarify this distinction and to differentiate between the first set of results using the base medium, single modifications and the rational design of combinations of these. We ensured that these terms were changed throughout the manuscript to be clearer for the reader. For example:

Line 84-87: “To enhance the recovery of fastidious gut microbes, we modified the medium’s composition by introducing hemin, vitamin K1, and antioxidants. This modified recipe was used as the basis for all experiments as is and is referred to as Base medium throughout the paper.”

Line 134-136: “Most diversity was exclusive to the original stool samples, with 149 mOTUs absent in single modification cultivations. Overall, our single modification cultivations recovered 42% of species that were detected in the stool samples (105 shared mOTUs).”

Line 276-278: “To select combinations of media modifications for the enrichment of a desired microbial function or taxa, we devised a method that leveraged taxonomic and functional targets together with the data collected from the metagenomes of our single modification cultivations (Figure 6A).”

I understand the flow of the narrative, but I think the specific meaning of “benchmark conditions” in this manuscript needs to be provided when first mentioned at line 126, even if the full explanation of the combinations is left until later. The title describes “metagenome-guided culturomics” but it appears that metagenomes were used primarily as a phylogenetic readout. The use of metagenomic information to guide the combination cultivations had exciting promise, but instead, nearly all analyses relied solely on extracted 16S fragments (mOTUs), and it’s not clear that any functional information or annotation was used. I would have liked to see more extensive use of the metagenome data, both for strain quantification and for functional interpretation, as is briefly alluded to in lines 360-367. The process used to score and select combinations needs to be much better explained. Ideally, software used for this would be shared with the community to allow others to develop their own media modifications based on the data.

We thank the reviewer for their thorough work in reviewing our manuscript and hope they find that the revised version has been significantly improved based on their feedback. We are happy to hear that the reviewer can see the potential of the work and have done our best to address the issues raised. We fully agree with the reviewer that is a big potential to use the metagenomic data to answer more questions, but have chosen to use the mOTUs database to perform taxonomic assignment as the basis for this manuscript, with the idea of making the data generated available for the scientific community allowing people to answer their question of interested with this resource. mOTUs database query leverages multiple marker genes in metagenomes in each sample to establish phylogenetic relatedness. While it is not as low resolution as 16s taxonomic assignment, it’s also not as accurate as fully resolved assembled genomes that would give a strain resolution. We also showcase a potential use of metagenomic data in figure 6D based on functional assignment to KEGG orthologous groups for Dopamine and DOPAC synthesis pathways. We further explained the process to select the combinations in the method section.

Comments:

3.1 Line 83: “To enhance the recovery of fastidious gut microbes, we modified the medium’s composition by introducing hemin, vitamin K1, and antioxidants.” Why would these enhance the recovery of fastidious gut microbes? Has this been shown before or was it just hypothesized?

It has been shown before that hemin, vitamin K1 and antioxidants are effective culture supplements used in growth media for anaerobic, fastidious microorganisms (Dione et al., 2002; Roe et al., 2002; Million et al., 2020).

References:

- Dione, N., Khelaifia, S., La Scola, B., Lagier, J. C., & Raoult, D. (2016). A quasi-universal medium to break the aerobic/anaerobic bacterial culture dichotomy in clinical microbiology. *Clinical Microbiology and Infection*, 22(1), 53–58. <https://doi.org/10.1016/j.cmi.2015.10.032>
- Roe, D. E., Finegold, S. M., Citron, D. M., Goldstein, E. J. C., Wexler, H. M., Rosenblatt, J. E., Cox, M. E., Jenkins, S. G., & Hecht, D. W. (2002). Multilaboratory comparison of anaerobe susceptibility results using 3 different agar media. *Clinical Infectious Diseases*, 35(Suppl 1), S40–S46. <https://doi.org/10.1086/341919>
- Million, M., Armstrong, N., Khelaifia, S., Guilhot, E., Richez, M., Lagier, J. C., Dubourg, G., Chabriere, E., & Raoult, D. (2020). The Antioxidants Glutathione, Ascorbic Acid and Uric Acid Maintain Butyrate Production by Human Gut Clostridia in The Presence of Oxygen In Vitro. *Scientific Reports*, 10(1). <https://doi.org/10.1038/s41598-020-64834-3>

3.2 Line 103: “The fifth group contains seven host-derived metabolites known to modulate the gut microbiota and host physiology: primary and secondary bile acids.” I found this phrasing a bit confusing – are all seven modifications bile acids? If so, perhaps “The fifth group contains seven primary and secondary bile acids, host-derived metabolites known to modulate the gut microbiota and host physiology” would be correct?

We agree with the reviewer that this is clearer and have updated the main text with the suggested correction (Line 105-107):

“The fifth group contains seven primary and secondary bile acids, host-derived metabolites known to modulate the gut microbiota and host physiology”

3.3 Figure 1: This figure leaves out the combinations entirely, which is especially confusing when Figure 2 includes results of the combination experiments before those are described in the text. At the same time, nothing is said in the legend about the parts of the figure labeled “Benchmark” and “Databases” which are hardly mentioned in the text. Could the figure be modified to include some indication of where the combinations fit in, or the title modified to note that it only covers the “benchmark” experiments?

We thank the reviewer for this suggestion. We have updated the figure to more accurately reflect the experimental workflow, and how the combination of modifications fit in. Based on the feedback from several reviewers, we have also moved the presentation of the results from the combination of modifications to figure 6A, and hope the story is clearer in the revised version.

3.4 Figure 1: The legend mentions 612 plates but doesn’t indicate how this number was reached. Based on my reading of the method it appears to be from 6 donors X 51 cultivation conditions (50 plus base) X 2 dilutions. I think this is worth spelling out, particularly the two dilutions and the lack of any replicates.

We have updated the figure legend to include the calculation (6 donors x 51 cultivation conditions (50 plus base) x 2 dilutions). We found that technical replicates was not feasible due to the sheer amount of plates and anaerobic workspace required for this.

Line 115-121: “Figure 1: Overview of the experimental design 50 growth medium modifications were selected for our initial single modification cultivations. Each additive or modification was applied to the base medium: a modified Gifu Anaerobic Medium (mGAM). Stool samples from six healthy donors were plated on each medium modification and incubated anaerobically. After

seven days, the 612 (6 donors x 51 conditions x 2 dilutions) plates were scraped to recover the biomass and prepared for sequencing on an Illumina NovaSeq.”

3.5 Line 131: “Notably, Firmicutes, mainly Clostridiales species incertae sedis, were only detected in our stool samples, possibly due to metagenome assembled genomes (MAGs).” I couldn’t understand what this was intended to say, especially since MAG assembly wasn’t part of the scope of the work in the manuscript. I assume the failure of these species to appear in the cultivations simply means they aren’t amenable to the cultivation conditions used.

Upon consideration, we agree that the original sentence may have caused confusion, particularly since MAGs were not a focus of this study. Therefore, to avoid any potential misunderstandings, we have removed this sentence from the manuscript. We appreciate your feedback in helping us clarify this point.

3.6 Line 136: “...our cultivation approach can support growth of diverse taxa that is inaccessible to cultivation-independent methods.” Since whole metagenome sequencing was used to characterise both the stool communities and the cultivation products, clearly these organisms are accessible to “cultivation-independent methods” like metagenomics, but are presumably too low in abundance in the original stool samples to be detected at the depth of sequencing used.

We agree with this point that was also raised by another reviewer, and have modified the statement to clarify on this matter (Line 137-140) :

“These findings, consistent with culturomics studies^{17,28}, indicated that our cultivation approach can support growth of diverse taxa that were undetected at the depth of sequencing used in our cultivation-independent surveys”

3.7 Figure 3: Are the values displayed here treating each dilution as a separate data point, or combining the results of the two dilutions for each donor/media combination?

Only one of the dilutions was used for any conditions, in each case the plate with the most biomass without a lawn of growth was used. We have updated the methods section under ‘Sample preparation for sequencing’ clarify on this matter (Line 466 – 489)

3.8 Line 159: “...PD variation was also noted among media modifications, particularly in histidine, chloramphenicol, vancomycin, CA, GCA, caffeine, ethanol, fluoride, nanoparticles, 10 dilution, and pH4, which were consistently beneficial across different groups.” I couldn’t see anything distinguishing about this list of modifications in Figure 3B – many, but not all, of them have 95% confidence intervals that cross over Faith PD index of 10, but their average PDs are not necessarily higher than some of the other modifications not on the list, like tryptophan, mucin, or isobutyrate. And does “beneficial” mean “increasing diversity” in this case? What exactly is the comparison to?

The variation in PD is significantly influenced by donor compositions. However, some media demonstrate higher PD than the base medium and are associated with increased diversity on plates, indicating they are less selective compared to media like GCA or 10X. If the objective is to reduce diversity, targeting enrichment towards specific taxa with higher relative abundance, other media with lower PD relative to the base may be more appropriate. These effects are more clearly observed in donor-specific plots, though six plots would be required, as with OTU richness

mentioned earlier. We have revised the language from “beneficial” to “associated with increased diversity,” acknowledging the reviewer's point that “beneficial” can be open to interpretation.

3.9 Line 164: “Relative abundance was dominated by Bacteroidetes or Proteobacteria across most modifications (Figure 3C).” I couldn’t discern any information about phylum-level relative abundance from Figure 3C, and in fact a much larger number of the families listed in that figure are Firmicutes than Proteobacteria. If phylum-level abundance is going to be referenced, perhaps this needs to be included in a supplemental figure.

No phylum-level relative abundance data are presented in the figure 3C. Nevertheless, the figure legend enables the reader to read the changes in relative abundance across media. We have restructured this sentence to clarify this (Line 166-169) and added Supplementary Figure 1 to justify the statement:

“Relative abundance was dominated by Bacteroidetes or Proteobacteria across most modifications (Supplementary Figure 1) with exceptions in pH5, ethanol, sodium salt, nanoparticles, caffeine, inulin, taurine, vancomycin, clindamycin, ciprofloxacin, chloramphenicol, and cefotaxime where a reduction in relative abundance for these families can be observed (Figure 3C).”

3.10 Line 168: “...suggesting a link between media modification, microbial community composition, and PD increase.” This seems like kind of an obvious statement, that media composition will affect community diversity and composition; was any deeper insight obtained?

Thank you for the feedback. The reviewer brings up a valid point regarding this statement. In response, we have revised the text to discuss specific changes in taxa (Lines 169-173):

“Interestingly, these media modifications also achieved the largest phylogenetic diversity, often from Firmicutes and Akkermansiaceae species, suggesting that these specific modifications can disrupt the dominance of more abundant taxa, such as Bacteroidetes or Proteobacteria, allowing less abundant taxa to proliferate.”

3.11 Line 170: This paragraph makes a lot of assertions about particular modifications leading to more or less phylogenetic diversity, referencing Figure 3C. Is this specifically family-level diversity (as compared to the PD index in 3B), and is there any statistical significance associated with these statements? Many of the statements here seemed anecdotal and given the lack of replication may not be reproducible.

No statistical tests were conducted on these observations. The observations were intended to be descriptive, highlighting patterns that may warrant further investigation.

3.12 Line 180: “Lachnospiraceae, Oscillospiraceae, and Ruminococcaceae species, known for health-promoting capabilities, were selectively enriched in multiple conditions...” I assume this is referencing Figure 3C, but what I understand Figure 3C displays is total family-level abundance of these groups. Did individual species within these families also consistently increase with these modifications, which isn’t quite the same thing since the family-level increase could be due to a smaller set of species? Also it merits mentioning that these three families displayed similar abundance profiles based on the clustering shown in Figure 3C, although the distance metric and clustering method used for that figure are not clearly stated as far as I can tell.

Thank you for your insightful comments. You are correct that Figure 3C shows the total family-level abundance of the Lachnospiraceae, Oscillospiraceae, and Ruminococcaceae. The increase at the family level may indeed reflect the contributions of a subset of species within these families, rather than a uniform increase across all species. The response of individual species can vary depending on the specific conditions and the diversity within each family, which might lead to different conclusions if analyzed at a finer taxonomic level.

We chose to present data at the family level because it provides a broad overview that is accessible to many microbiologists, facilitating a general understanding of the study's effects. However, we encourage readers to delve into the species-level data if their research focus requires a more detailed analysis. Regarding the clustering in Figure 3C, we acknowledge that the distance metric and clustering method were not explicitly stated, and we will clarify these details to enhance transparency.

3.13 Line 183: “Similarly, ethanol treatment, caffeine, nanoparticles and fluoride are also enriched for these taxa.” Does “taxa” refer to just Lachnospiraceae or all three families mentioned earlier in the paragraph? And how does this add to the first sentence, which already mentions caffeine, ethanol, and fluoride as enriching for the three families?

Thank you for pointing that out. We have clarified the statement with Line 188-189:

"Similarly, nanoparticles appear to better support the growth of Oscillospiraceae"

3.14 Line 187: Here, as in line 168, “suggesting” doesn’t seem like the right phrasing as donor and media are clearly expected to influence taxonomic composition.

Thank you for pointing that out. We have rephrased it using the term "indicating," which we believe is more appropriate.

3.15 Line 199: “A principal component analysis (PCA) revealed 40.8% variance could be explained by the first two axes (Figure 4A), highlighting that the baseline composition of donors had a strong influence over the distance between samples.” For me this doesn’t follow – the axes have no specific relationship to the donors so anything could be driving the differences between communities based on this display. I think some other statistical test is needed to make this point, like the permanova test mentioned later in the paragraph.

The reviewer insight is correct. The PCA is used as a basis to form our assumption that donor composition might influence the distance between samples. This is the reason why other statistical tests are employed. What the PCA plot aims also to show is that most samples for each donor tend to cluster around the base media for that donor (now highlighted by a black circle). We reformulated the paragraph to make this clearer in line 202-210:

“The Hellinger distance was used to assess the degree of similarity between media modifications and donors. A principal component analysis (PCA) revealed 40.8% variance could be explained by the first two axes (Figure 4A), highlighting that the initial composition of donors had a strong influence over the distance between samples. To further characterize the relationship between donors, a Mantel test was used, showing significant correlations between the donor distance matrices (Figure 4B). Finally, PERMANOVA analysis confirmed the significant impact of donor composition on genus-level microbiome composition ($F=20.55$, $p<0.001$) without affecting sample dispersion (PERMDISP: $F=1.43$, $p=0.204$, 999 permutations). These results indicate that the baseline donor composition substantially influences the microbiome composition across most medium modifications.”

3.16 Line 201: “a Mantel test was used, showing significant correlations between the donor distance matrices (Figure 4B).” This sentence was meaningless to me. What is that test assessing and what is a “donor distance matrix”? Is this just showing that there are trends in relative abundance that are consistent across gut samples? Is Figure 4B based on stool profiles only, not cultivated communities?

We thank the reviewer for pointing this out. We have updated the figure legend to clarify how the analysis was performed. The mantel test is performed on the sequencing extracted from cultivation data. In line 234 – 240

“B. Heatmap of the Mantel test results between donor distance matrices using the Pearson R correlation coefficient (p -value <0.01 for all correlation coefficients). The Hellinger distance between modifications at genus level is calculated for each donor, these distance matrices are leveraged in a mantel test to calculate correlations between donors. A positive correlation means that there is a correlation between distance matrices for a set of donors, and that conditions tend to enrich for the same genus in a particular medium.”

3.17 Line 206: “Nonetheless, some modifications also exhibited a significant effect on composition at genus level ..., with a subset of samples demonstrating taxonomic diversity beyond the donor baseline (Figure 4A, marked in lavender).” How are the lavender circles in 4A calculated and which points (if any) correspond to the “donor baseline” (which I assume is the stool sample profile)? They appear to be arbitrarily drawn at the edges of the plot, and without the plots being annotated with modifications the next sentence, “Interestingly, most of those samples are associated with a more diverse taxonomic enrichment compared to base medium (Figure 3B and C),” cannot be assessed.

We thank the reviewer for the comment. We reformulated the paragraph to clarify the message. We hope the reviewer will find these overall changes satisfying. In line 209-221:

“These results indicate that the baseline donor composition substantially influences the microbiome composition across most medium modifications. However, modifications also demonstrated a significant effect on genus-level composition (PERMANOVA: $F=2.42$, $p<0.001$) without affecting sample dispersion (PERMDISP: $F=1.01$, $p=0.189$, 999 permutations), with some samples exhibiting taxonomic diversity beyond their initial donor composition (Figure 4A, marked in lavender). Overall, the average ratio of within-donor variance to between-modification variance (Figure 4C and Methods) was below 0.1. This indicates that the genus relative abundance variation due to modifications was, on average, ten times greater than the variation attributed to differences between donors. Some modifications, such as caffeine, resulted in different taxonomic diversity compared to the base medium (Figure 3B/C) but showed a stronger influence from the initial donor composition (higher ratio value in Figure 4C). In contrast, other modifications, such as CDCA, had a more pronounced influence on taxonomy relative to donor influence (lower ratio) and exhibited lower diversity.”

3.18 Line 213: What does “superior” mean in the context of abundance variation – greater or less?

We appreciate this comment. We did not realize the term was unclear, therefore we have revised the text to clarify that it is ‘larger’.

3.19 Line 214: “...although donor influence was in general the major contributor to final taxonomic composition observed, certain modifications exerted a notable impact at the genus level.” Which data demonstrate this and what is the statistical significance, particularly the next

statement that some modifications had “a more substantial influence than the donor effect”?
We appreciate the opportunity to clarify this point. The data supporting the statement regarding donor influence versus modifications can be found in Figure 3 and in discussed in detail in line 202-210:

“The Hellinger distance was used to assess the degree of similarity between media modifications and donors. A principal component analysis (PCA) revealed 40.8% variance could be explained by the first two axes (Figure 4A), highlighting that the initial composition of donors had a strong influence over the distance between samples. To further characterize the relationship between donors, a Mantel test was used, showing significant correlations between the donor distance matrices (Figure 4B). Finally, PERMANOVA analysis confirmed the significant impact of donor composition on genus-level microbiome composition ($F=20.55$, $p<0.001$) without affecting sample dispersion (PERMDISP: $F=1.43$, $p=0.204$, 999 permutations). These results indicate that the baseline donor composition substantially influences the microbiome composition across most medium modifications.”

3.20 Lines 237-238: Bacteroides and Proteobacteria are twice referred to as “genera” but Proteobacteria is a phylum. Based on Figure 5B I think the reference is actually to an unnamed genus within the Proteobacteria but that needs to be clarified. And I couldn’t see any data supporting the statement that “these two genera are usually present and dominant in most modifications, while other genera vary more broadly in occurrence” since no genus-level abundance data are presented. Similarly, the next sentence, “Interestingly, we observed a clustering of Firmicutes and Actinobacteria, revealing potential similar media preferences,” lacks a figure reference and doesn’t appear to be supported by Figure 5B, where genera within the Firmicutes, Bacteroidetes, Proteobacteria and other phyla are intermingled in the tree with the relatively few Actinobacteria genera. I think all of these statements need better support if they are included, but they also need some biological interpretation to be useful and the current Discussion lacks any discussion of phyla or genera and their media preferences.

We thank the reviewer for pointing this lack of clarity. We believe that the statement regarding the proximity of Firmicutes and Actinobacteria in the clustering is supported by both the two clusters visible in the figure (Proteobacteria and Bacteroides in one blue cluster and the other genera in a brown cluster) and their proximity within subclusters both globally and in specific cases (e.g., Intestinibacter and Brevibacterium). We have updated the text to better align with the data presented. Lines 254-260:

“Genus-level visualization of the growth profile distance in media modifications also confirmed this observation (Figure 5B), highlighting a segregation of Bacteroides and genus belonging to the Proteobacteria phylum. This is consistent with the phylum level data (Supplementary Figure 1) where Bacteroides and Proteobacteria are usually present and dominant in most modifications, while other genera vary more broadly in occurrence. Interestingly, we observed a clustering of genus belonging to Firmicutes and Actinobacteria, revealing potential similar media preferences.”

3.21 Figure 5A: It would help to have some explanation of how this plot was generated. Is this just a standard boxplot, but with individual data points plotted on the right hand side?

Thank you for your feedback on Figure 5A. You are correct in your understanding: the plot is indeed a standard boxplot with individual data points displayed on the right side. This additional

layer allows for a clearer visualization of the distribution and variability of the data points. We hope this explanation clarifies the figure's presentation.

3.22 Line 258: “We calculated and ranked combinations based on mean relative abundance and absolute count for targets across modifications for each donor and each combination.” A better explanation is needed for how combinations were scored and ranked. As far as I can tell the scoring and ranking process is not described in the methods. An attempt is made in Figure 6A but it isn't understandable and definitely wouldn't be reproducible.

We thank the reviewer for pointing this out. We have updated the methods to better explain how the figure was generated. Lines 497-512:

*“Functional assignment to KO groups was used to design media for functional targets while taxonomic profiles were used for the design of media towards specific taxa such as *Collinsella aerofaciens*, following the process described in Figure 6A. Briefly, the design of targeted media was done in 4 steps. First, all possible combinations of media were generated for the conditions used in the single modification cultivations, up to 6 conditions per combination, amounting to 10.9M combinations. Then, metrics were calculated to score what could be the most likely combinations to yield an enrichment for the target: Mean relative abundance per OTU/function, total number of OTUs/functions in the combination and ratio of the mean relative abundance for the target over the sum of the mean relative abundance for all OTUs/Functions in the combinations. Finally, combinations were ranked for each target based on highest Ratio > lowest total number of OTUs/function > highest mean relative abundance of the target and the top 20 modifications for all the combinations for a target were retained. Multiple visualizations were generated to assist in choosing the combinations to test: co-occurrence network of the most present modifications in the best combinations across donors, heatmap of the best modifications relative abundance per donor and the mean relative abundance of the target relative to the number of OTUs/functions in the combination.”*

3.23 Line 268: It would be helpful to clearly state that *Collinsella* relative abundance is the metric being used here for improvement.

We agree and have added this to clarify. (Line 288-290):

*“This combination of medium modifications yielded a significant improvement of 10 to 100-fold increase in the relative abundance of *Collinsella aerofaciens* (From 0.015% to 2%) over the single modification cultivation”.*

3.24 Line 275: “A combination of 4 media modifications (acetate, pectin, 10X dilution, pH6) was able to achieve 10 to 20-fold enrichment in dopamine and DOPAC pathways over the mean relative abundance in the base medium.” This implies that these pathways were directly assessed in the metagenome data – if so how? The Figure 5C legend says “Bar plot for the fold change enrichment over Base for strains associated with dopamine and DOPAC synthetic pathways,” which implies that mOTUs were used as proxies for these activities. Again, no details are provided in the methods on how these pathways were assessed and how combinations of modifications were predicted and tested. This needs to be much more clearly explained.

We thank the reviewer for pointing this out. We have updated the methods to better explain how the figure was generated. Lines 497-512.

3.25 Figure 6D is very hard to follow.

We thank the reviewer for pointing this out. We have updated the legend to better explain how the figure can be read. Line 331-346:

*“Figure 6: Combinations of conditions further improve the targeted enrichment of taxa and functions of interest. A. Overview of selection process of modification combinations. Depending on the targets (microbial functions or taxa), different metrics can be calculated, in order to achieve the maximum likelihood for enrichment, based on the co-occurrence of targets in the different modifications (see Methods). B. Venn diagram of the OTUs detected in stool, single modification, and combinations. C. Bar plot for the enrichment in relative abundance (%) for *Collinsella aerofaciens*. Single modification cultivations are colored in green and combinations in orange. D. Bar plot for the fold change enrichment over Base for strains associated with dopamine and DOPAC synthetic pathways, based on functional assignment to KEGG orthologous groups for all samples. E. Proportion of species or functions present in different modifications composing the combinations, for different numbers of modifications per combination. Each bar represents the proportion of species or functions detected in the specified number of modifications. For example, in combinations composed of 4 modifications, close to 40% (orange) of the species are detected in all 4 modifications, while around 30% (dark blue) are not detected in any single modification composing the combinations. F. Fold change enrichment over Base in logarithmic scale for *Bifidobacterium* and *Lactobacillus* species in MRS and ISO-0019.”*

3.26 Figure 6E is so briefly explained it's hard to grasp its purpose – how was this medium designed and what was the intended outcome? The methods do not include an explanation.

*We thank the reviewer for pointing this out. The Figure 6F aims to showcase that even when media are not rationally designed for a particular target (taxonomic here instead of functional), valuable outcomes can occur. Indeed, this goal was not rationally designed for *Bifidobacteria*, but designed for formate metabolism, which is present in *Bifidobacteria* and *Lactobacillus* metabolism. While not targeting a particular taxon, the medium shows to be beneficial for the recovery of a particular group that carry these pathways. We updated the text to highlight this point in line 309-312:*

*“We also observed certain combinations that outperformed conventional media. A combination of lactate, mucin, fluoride, nanoparticle and 10X dilution showed markedly increased enrichment of *Lactobacilli* and *Bifidobacteria* compared to the gold-standard MRS medium (Figure 6F). This medium combination was initially designed for targeted bacteria with formate metabolism rather than taxa.”*

REVIEWER COMMENTS

Reviewer #2 (Remarks to the Author):

I thank the authors for the time and effort they took to reply to my comments and revise their manuscript. This is much appreciated. The following comments are intended to clarify some of my earlier points and to offer a few more suggestions, which I hope the authors will find useful.

(1) Figure 3B, data missing for rightmost two categories “total on media & stool” and “total on media only”: The authors reply that it would not be meaningful to calculate PD for these two categories. Still, I do not understand. PD is a measure that is calculated on each sample, in the same way mOTU richness is calculated on each sample in Figure 3A. In panel 3A, the points are shaded by donor, while in 3B the points are shaded by modification type. The authors have made a color for “stool sequencing”, why can't they make a color for the last two categories? Relatedly, the Results text concerning Figure 3B refers to something called “PD variation”; for example, “[o]verall, the highest PD variation stemmed from inter-donor initial composition.” But the variation in PD among donor stool samples – presumably depicted by the size of the rightmost boxplot – is certainly not the largest of those shown. The figure caption says PD is calculated “across modifications for all donors”, which doesn't really explain what each point is. And PD is not mentioned at all in the Methods. Altogether, I remain worried there's been some confusion, miscommunication, or mislabeling with respect to Figure 3B.

Response:

We appreciate the reviewer's detailed feedback and the opportunity to clarify our methodology and data presentation in Figure 3. In Figure 3A, we present species richness for each donor across different media modifications. We also include cumulative counts for 'total on media only' and 'total on media & stool' samples. Calculating total species richness for combined samples is meaningful because we aim at 1. providing insights into the overall taxonomic diversity captured from each donor throughout the study and 2. help evaluate how well our cultivation methods recovered the donors' microbiota diversity compared to the original stool samples. Species richness is a simple count of unique taxa and does not rely on phylogenetic relationships or assume ecological interactions among taxa.

Conversely, Faith's PD is designed as an alpha diversity metric that measures the phylogenetic diversity within a single community or sample by summing the branch lengths connecting all taxa present in that specific sample. Calculating PD for combined datasets that aggregate taxa from multiple distinct samples is not appropriate for several reasons. Firstly, Faith's PD calculation assumes that the taxa are part of a single, cohesive ecological community where phylogenetic relationships are meaningful in the context of that community. Combining taxa from different samples violates this assumption, making such PD calculations statistically invalid and their interpretation unreliable. Secondly, aggregating taxa from multiple samples increases the total number of observed taxa and the summed branch lengths, leading to artificially inflated PD values. Such inflated PD

values are not directly comparable to the PD of individual samples, as they represent a different level of diversity (gamma diversity rather than alpha diversity), and could mislead readers about the diversity of individual communities.

All in all, because species richness is a simple count of unique taxa without considering phylogenetic relationships or community structure, summing species richness across samples provides an aggregate measure of total diversity, which is informative for assessing overall taxonomic coverage. Conversely, Faith's PD incorporates evolutionary relationships among taxa within a specific community, requiring taxa to be part of a coherent ecological unit for phylogenetic relationships to be meaningful.

We acknowledge that the Methods section and figure legend lacked details on PD calculation. We have updated it to provide a clear explanation:

Methods: *“Faith’s Phylogenetic Diversity (PD) was calculated for each individual sample using the observed mOTUs and their phylogenetic relationships. A comprehensive phylogenetic tree was constructed using all mOTUs detected across samples, based on the alignment of conserved marker genes from the mOTU database. For each sample, PD was calculated by summing the branch lengths connecting the taxa present in that specific sample using the ‘faith_pd’ function from the scikit-bio 0.5 library. No PD calculations were performed for “Total on media and stool” and “Total on media only” as those calculations would not be performed on a single cohesive ecological community where phylogenetic relationships are meaningful in the context of that community.”*

“Figure 3B: Faith’s Phylogenetic Diversity (PD) index calculated for each individual sample across all medium modifications and donors. Each point represents the PD of a specific medium modification for a specific donor and is grouped by modification type and colored accordingly. Boxplots summarize the distribution of PD values within each modification group”

We have clarified the discussion of PD in the Results section line 161:

“The original stool samples exhibited higher phylogenetic diversity (PD) that corresponded to increased mOTU richness (Figure 3B). Overall, while some variation in PD was observed across donors due to differences in inter-donor initial composition, the media modifications had a marked influence in driving PD changes. In particular, conditions such as histidine, chloramphenicol, vancomycin, cholic acid (CA), glycocholic acid (GCA), caffeine, ethanol, fluoride, nanoparticles, 10X dilution, and pH4 were consistently associated with increased PD across different donors. Conversely, treatments like clindamycin, tetracycline, chenodeoxycholic acid (CDCA), deoxycholic acid (DCA), taurocholic acid (TCA), sodium salts, and aerobic incubation resulted in the lowest PD, indicating that these conditions led to a selection of fewer, more phylogenetically similar OTUs”

(2) Figure 4A, hand-drawn ovals: I understand better now the purpose of the hand-drawn ovals, but I think the figure caption should further specify that an arbitrary number of samples appear within each hand-drawn oval. (In fact, a few samples are half in, half out.) The legend should also probably note that hand-drawn ovals don't highlight any particular type of modification.

Response:

We thank the reviewer for this comment and appreciate the suggestion to clarify the role of the hand-drawn ovals in the figure legend. We have now revised the legend for Figure 4B to explicitly state that the hand-drawn ovals are used to highlight clusters of samples located at the extremities of the PCA plot. These ovals are meant to indicate areas where multiple donors share a similar biological response across conditions, without implying any specific biological modification or grouping. We have also noted that the number of samples within each oval is arbitrary, and some samples may appear partially inside or outside the ovals.

We hope this revision helps clarify the visual representation of the data and aligns with the reviewer's recommendation.

(3) Figure 4B, Mantel tests: I appreciate the authors confirming I guessed correctly about the inner workings of their Mantel test. But I don't think the results show that conditions tend to enrich for the same genus in a particular medium; a distance is a distance, not a direction. Donors 1 and 2 might each be pushed far by modification A, but it could be that modification A selects for completely different genera in donor 1 (versus donor 1 base) than in donor 2 (versus donor 2 base).

It seems like what you'd like to visualize within each modification is (1) the average of the pairwise distances between different donors, versus (2) the average of the pairwise distances between each donor and their own base. [It just seems like you're trying to identify modifications in which donor composition moves far (high #2) AND in the "same (compositional) direction" (low #1).] I suspect this is what Figure 4C is sort-of getting at, but the writing and labeling around 4C remain difficult for me to parse.

Response:

We appreciate the reviewer's insightful feedback regarding our use of the Mantel test in Figure 4B and the interpretation of our results. Your comments have highlighted important considerations about the limitations of the Mantel test and have prompted us to refine our explanations to avoid overinterpretation. We agree that the Mantel test measures the correlation between distance matrices, providing information about the similarity in the patterns of community dissimilarities between donors across media modifications. Specifically, it assesses whether donors exhibit similar relationships in how their microbial communities respond to different modifications in terms of distance. However, as you point

out, the Mantel test does not provide information on the direction of change or the specific taxa that are enriched or depleted in response to the modifications. Our original phrasing may have inadvertently suggested that the Mantel test results indicate that the same genera are enriched across donors for a particular medium modification, which is not accurate. We acknowledge that donors could experience similar magnitudes of change (distances from their baselines) without necessarily converging toward the same microbial compositions.

To address this, we have revised the text in the Results section to more accurately reflect the interpretation of the Mantel test results and to clarify our conclusions.

Original : *"To further characterize the relationship between donors, a Mantel test was used, showing significant correlations between the donor distance matrices (Figure 4B)."*

Revised: *"To further characterize the relationship between donors, we performed a Mantel test to assess the correlation between the patterns of community dissimilarities (as measured by Hellinger distances at the genus level) across donors (Figure 4B). Significant positive correlations indicate that the overall patterns of how media modifications affect community composition are similar between donors. This suggests that modifications causing large shifts in one donor's community tend to cause large shifts in others as well, although not necessarily involving the same genera."*

Regarding Figure 4C, we acknowledge that the explanation may have been unclear, and we appreciate the opportunity to clarify its purpose. Figure 4C illustrates the ratio of within-donor variance to between-modification variance for each medium modification. This ratio helps quantify the relative influence of donor-specific effects versus modification effects on the genus-level microbial community composition. We have revised the text to better explain this concept and connect it to other results from the Figure 4 in the Result section (line 227):

"To quantify the relative influence of donor composition and media modifications, we calculated the ratio of within-donor variance to between-modification variance for each medium modification (Figure 4C and Methods). A lower ratio indicates that the variation in genus composition is more strongly influenced by the modification than by donor-specific differences, suggesting that the modification tends to drive microbial communities of different donors toward more similar compositions. Conversely, a higher ratio suggests that donor composition has a stronger influence, and the modification's effects vary among donors. Overall, the average ratio was below 0.1, indicating that the genus relative abundance variation due to modifications was, on average, ten times greater than the variation attributed to differences between donors. Some modifications, such as caffeine, resulted in different taxonomic diversity compared to the base medium (Figures 3B and 3C) but had higher variance ratios (Figure 4C), indicating a stronger influence from the initial donor composition. In contrast, other modifications, such as CDCA, had lower variance ratios, suggesting a more pronounced influence of the modification on taxonomy relative to donor influence and exhibited lower diversity."

(4) Figure 5A, samples or taxa: When I first viewed this plot, I assumed that each point represented a Hellinger distance between a pair of samples, and that taxa were glommed at the given taxonomic level prior to calculation of the sample-to-sample distances. But now, noticing that Phylum has many fewer points (and that my prior interpretation wouldn't make much sense given the question posed by the authors), I realize that maybe points represent Hellinger distances between pairs of taxa. In either case, there's an awful lot of (non-independent) observations here. Perhaps the authors could display for each taxon(?) the average of its distances to all others? In general, I would encourage the authors to explain in greater detail the entities and comparisons displayed.

Response:

We thank the reviewer for pointing out the lack of clarity in the figure explanation, and we apologize for any confusion caused. We agree that the figure legend and associated text could be improved to better convey the entities and comparisons displayed.

Figure 5A illustrates the relationship between phylogenetic relatedness and similarity in growth profiles across different media modifications. Each point represents the Hellinger distance between the growth profiles of a pair of taxa at a specific taxonomic level (Phylum, Family, Genus, Species). For each taxon, we constructed a growth profile based on its relative abundance across all media modifications, reflecting how frequently and abundantly the taxon is detected in each modification. We calculated the Hellinger distances between the growth profiles of all possible pairs of taxa within each taxonomic level. This results in fewer comparisons at higher taxonomic levels (e.g., Phylum) and more comparisons at lower levels (e.g., Species). The purpose of this analysis is to assess whether more closely related taxa exhibit more similar growth profiles. The decreasing median Hellinger distance from phylum to species level indicates that closely related taxa tend to have more similar growth responses across media modifications, suggesting a phylogenetic influence on growth preferences.

We acknowledge the reviewer's concern about the large number of pairwise comparisons and the potential non-independence of observations. We considered summarizing the data by displaying, for each taxon, the average of its distances to all others, as suggested. However, we believe that presenting the full distribution of pairwise distances provides valuable insights into the variability and overlap of growth profile similarities among taxa. Displaying only average distances might obscure important details about the range and distribution of similarities, especially in cases where certain taxa have particularly similar or dissimilar growth profiles compared to others.

To enhance clarity, we have updated the figure legend and the manuscript text to provide a more detailed explanation of the entities and comparisons displayed in Figure 5A. (line 295):

"A. Distribution of pairwise Hellinger distances between taxa at different taxonomic levels (Phylum, Family, Genus, Species). Each point represents the Hellinger distance between

the growth profiles of a pair of taxa at the specified taxonomic level. The decreasing median distances from phylum to species level indicate that more closely related taxa have more similar growth profiles, suggesting a phylogenetic influence on growth preferences."

Line 272: "We investigated whether growth in media modifications is associated with phylogeny by analyzing the similarity of growth profiles between taxa at different taxonomic levels (Figure 5A). For each taxon, we constructed a growth profile based on its relative abundance across all media modifications. We then calculated the Hellinger distances between all possible pairs of taxa within each taxonomic level. The median pairwise Hellinger distance decreased from phylum to species level, indicating that more closely related taxa tend to have more similar growth profiles. This suggests that phylogenetic relatedness influences the response of taxa to different media modifications. Genus-level visualization of the growth profile distance in media modifications also confirmed this observation (Figure 5B), highlighting a segregation of Bacteroides and genus belonging to the Proteobacteria phylum."

We appreciate the reviewer's feedback, which has helped us improve the clarity of the figure and manuscript. We have updated the figure legend and the text to provide more detailed explanations of the entities and comparisons displayed in Figure 5A. We believe that these revisions address the reviewer's concerns and enhance the reader's understanding of our analysis.

(5) Figure 6CD, relative abundance versus fold change: Okay, it just seemed odd to switch from one (panel C; rel. abund.) to the other (panel D; fold change). By the way, it would be helpful to edit the panel D title to something like "Dopamine & DOPAC biosynthetic gene cluster", or similar.

Response:

The choice to present relative abundance in panel C and fold change in panel D was made deliberately, as we believe it better highlights the distinct points being conveyed by each panel. In panel C, relative abundance emphasizes the proportional representation of specific taxa across conditions, while fold change in panel D more effectively illustrates the degree of enrichment for dopamine and DOPAC biosynthetic pathways. This distinction allows for a clearer interpretation of how these media modifications influence specific functional outcomes.

Additionally, we have updated the title of panel D to: "*Dopamine & DOPAC biosynthetic gene cluster*" as suggested, to improve clarity.

(6) Line 311, Lactobacilli and Bifidobacteria: Here and throughout, the authors are encouraged to review the formatting of informal names versus genus names, etc. For example, here the genus names are *Lactobacillus* and *Bifidobacterium* (italicized), while the informal names are lactobacilli and bifidobacteria (non-italicized).

Response:

Thank you for pointing this out. We have reviewed the formatting of genus and informal names throughout the manuscript. In instances where we refer to the genera, we have ensured that they are italicized, while the informal names "lactobacilli" and "bifidobacteria" are presented in non-italicized format.

(7) Line 471, "without obtaining a loan": I believe the authors intend the word "lawn" here, rather than "loan".

Response:

We thank the reviewer for bringing this to our attention, and have corrected this typo.

(8) Availability of data and code: Thank you for providing a reviewer link to your sequence data submission. The authors are further encouraged to make their analysis code available as well.

Response:

Based on the reviewer's recommendation, we have created a github repository to share code relevant to the understanding of the paper by the audience. This includes metadata or notebooks giving a more detailed explanation on how the combinations were designed for enrichment of particular targets, as well as code snippets used during the exploratory and analysis phase for generating the figures.

"All sequencing data was deposited in SRA database (Bioproject accession number: PRJNA1077691, <https://dataview.ncbi.nlm.nih.gov/object/PRJNA1077691?reviewer=kkk8qgid6jue1me3j1c4ddijje>). Metadata for the sequencing data and code related to the manuscript can be found on github at <https://github.com/Jerarm/Metagenome-guided-culturomics-for-the-targeted-enrichment-of-gut-microbes>."

The github repository is mentioned in the body of the paper as well in the method section. In addition, we remain available to answer any particular request that might come from readers.

Reviewer #3 (Remarks to the Author):

I have gone through the authors' responses and the revised manuscript and many of my concerns have been addressed, but several have not and need further work. Below are my remaining points that haven't been fully addressed:

Line 84 – In my previous review I asked why hemin, vitamin K1, and antioxidants were expected to enhance the recovery of gut microbes, and the authors responded with multiple citations. Yet the manuscript itself remains unchanged and these references have not been added – I assume

other readers might have the same question and I don't see why the authors wouldn't want to clarify this point.

Response:

We thank the reviewer for bringing this to our attention, and agree that other readers may indeed have the same question. We have restructured the sentence and included the relevant references:

"We modified the medium's composition by introducing hemin, vitamin K1, and antioxidants to enhance the recovery of fastidious gut microbes (Roe et al., 2002; Dione et al., 2016; Million et al., 2020)."

Figure 1 legend – In response to my question about the number of plates examined, the legend was modified to read, "After seven days, the 612 (6 donors x 51 conditions x 2 dilutions) plates were scraped to recover..." Yet elsewhere in the manuscript it is noted, "Only one dilution plate out of the two created for each sample was scraped, leaving a single dilution per sample to prepare for sequencing." Doesn't this mean only 306 plates were actually scraped?

Response:

We thank the reviewer for pointing out this error. We corrected accordingly the legend of Figure 1 to align with the Method section:

"50 growth medium modifications were selected for our initial single modification cultivations. Each additive or modification was applied to the base medium: a modified Gifu Anaerobic Medium (mGAM). Stool samples from six healthy donors were plated on each medium modification using two dilutions and 612 (6 donors x 51 conditions x 2 dilutions) plates were incubated anaerobically. After seven days, one dilution plate out of the two created for each sample was scraped to recover the biomass and prepared for sequencing on an Illumina NovaSeq."

Lines 166-173: the authors have added Supplementary Figure 1, addressing my concern that the statement about Bacteroidetes or Proteobacteria dominating most modifications wasn't supported by the data presentation. However, the sentence still reads "Relative abundance was dominated by Bacteroidetes or Proteobacteria across most modifications (Supplementary Figure 1) with exceptions in pH5, ethanol, sodium salt, nanoparticles, caffeine, inulin, taurine, vancomycin, clindamycin, ciprofloxacin, chloramphenicol, and cefotaxime where a reduction in relative abundance for these families can be observed (Figure 3C)" Part of my concern was that this sentence appears to refer to the phyla Bacteroidetes and Proteobacteria as families, but I think the authors intend to say that families within these phyla decrease in relative abundance so the sentence needs to be rephrased.

Response:

We appreciate the reviewer bringing this to our attention and apologize for the confusion caused by the incorrect use of taxonomic terminology in that sentence. We have revised the sentence to accurately reflect the taxonomic levels and clarify our findings (line 170):

"Relative abundance was dominated by members of the phyla Bacteroidetes and Proteobacteria across most modifications (Supplementary Figure 1), with exceptions in modifications such as pH5, ethanol, sodium salt, nanoparticles, caffeine, inulin, taurine, vancomycin, clindamycin, ciprofloxacin, chloramphenicol, and cefotaxime, where a reduction in relative abundance of families within these phyla was observed (Figure 3C)."

Lines 174-184: In response to my question about statistical significance, the authors acknowledged, "No statistical tests were conducted on these observations. The observations were intended to be descriptive, highlighting patterns that may warrant further investigation." Yet the phrasing of this paragraph remains unchanged and is still phrased as though these are well supported (e.g. "histidine and tryptophan enriched for more diverse taxa than tyrosine" and "taxa belonging to Eggerthellaceae, Clostridiaceae, Bifidobacteriaceae were positively enriched in CA, GCA and TCA cultures"). I would think that the six donors would provide sufficient sample numbers for statistical tests (for example, were those families consistently enriched in those treatments for all donors) but if these observations are not statistically supported they should either be removed or clearly noted as merely observational.

Response:

We appreciate the reviewer's valuable feedback regarding the phrasing of our observations. We acknowledge that the original wording may have inadvertently implied definitive conclusions without statistical support. Given the limited number of donors (n=6) and the inherent variability in human microbiome samples, we agree that statistical tests may not provide robust significance for these specific observations. To address the reviewer's concern, we have revised the paragraph to better indicate that these findings are preliminary and observational in nature (line 178). We believe that highlighting these patterns can provide valuable insights and guide future research efforts that include larger cohorts and more comprehensive statistical analyses.

"In the aromatic amino acids group, we observed that histidine and tryptophan tended to enrich for more diverse taxa than tyrosine, with tryptophan appearing to selectively favor Clostridiaceae, Bifidobacteriaceae, and Akkermansiaceae species compared to the base medium (as suggested by PD values in Figure 3B and the number of green squares in Figure 3C). Antibiotics generally resulted in lower phylogenetic diversity, particularly clindamycin and tetracycline, which detected on average three and four families, respectively (Figure 3C). Interestingly, imipenem, niclosamide, and piperacillin, despite different modes of action, showed similar enrichment profiles. Furthermore, taxa belonging to Eggerthellaceae, Clostridiaceae, and Bifidobacteriaceae seemed to be positively influenced in cultures containing CA, GCA, and TCA. SCFAs modifications showed similar

profiles at the family level, with some notable observations, such as a possible enrichment for Clostridiaceae in butyrate or Eggerthellaceae in formate."

Line 185: "Lachnospiraceae, Oscillospiraceae, and Ruminococcaceae species ... were selectively enriched in multiple conditions..." In response to my query about this statement relative to the data presented in figure 3C, the authors acknowledged that only family-level abundances were presented and shown to have these patterns, yet they failed to remove this reference to species that implies that individual species showed these patterns.

Response:

We appreciate the reviewer's attention to this detail. Upon review, we agree that referring to "species" in this context is inaccurate, as only family-level abundances were presented. We have now revised the text to state:

"Lachnospiraceae, Oscillospiraceae, and Ruminococcaceae families were selectively enriched in multiple conditions," to better reflect the data shown in Figure 3C. This correction ensures consistency between the results presented and the text.

The authors also said in their response, "Regarding the clustering in Figure 3C, we acknowledge that the distance metric and clustering method were not explicitly stated, and we will clarify these details to enhance transparency." As far as I can tell, this was not done.

Response:

We apologize for the lack of clarity. We have now improved this by correcting the figure legend to include a statement on the clustering method and distance metric used for generating the heatmap (line 205) :

"C. Heatmap of the mean Log10-fold change in relative abundance across donors for taxonomic families detected in single modification media modifications over GAM base. 0.001% was used as a baseline count for the calculations. Families are hierarchically clustered using Euclidean distance and the average linkage method to group families with similar abundance profiles across media modifications."

Line 191: In my previous review I noted that "suggesting" was not the best term for something that seemed very clear, and the authors responded "We have rephrased it using the term 'indicating,' which we believe is more appropriate." This also appears not to have been done since this line still reads "...suggesting media modification and donor-dependent effects on taxonomic composition."

Response:

We apologize for this omission and have implemented the correction.

Line 203: I and another reviewer both noted that the percent of variance explained by the PCA axes provides no insight into the factors driving the differences and the authors agreed. But this line still reads, “A principal component analysis (PCA) revealed 40.8% variance could be explained by the first two axes, highlighting that the initial composition of donors had a strong influence...” I think simply changing the word “highlighting” to “and highlighted that...” would clarify that this is apparent from the PCA itself, not from the percent variation explained by the axes.

Response:

We appreciate the reviewer’s careful attention to this point. We agree that the original wording could imply that the percent of variance explained by the PCA axes itself indicates the donor composition’s influence, which is not the case. Rather, it is the clustering pattern observed in the PCA that reveals the strong influence of donor composition. To clarify this, we have revised the sentence to read:

“A principal component analysis (PCA) revealed that 40.8% of the variance could be explained by the first two axes, and highlighted that the initial composition of donors had a strong influence on the separation of samples.”

This modification should better reflect the relationship between the PCA results and donor composition.

Line 214 and Figure 4C: “...the average ratio of within-donor variance to between-modification variance (Figure 4C and Methods) was below 0.1.” The y axis on Figure 4C is labeled “Variance between donors over variance between genera” which to me is meaningless; this is also in the legend (“Ratio of the variance within donors to the variance between genera for each modification”) but I think “genera” should be “modification” in both these places.

Response:

We thank the reviewer for pointing out this inconsistency. Upon review, we agree that the term “genera” in both the y-axis label and the figure legend should indeed be “modification,” as the intended comparison is between the variance within donors and the variance between modifications. We have corrected this in both Figure 4C and its legend to ensure clarity and consistency. The revised y-axis label now reads “Variance between donors over variance between modifications,” and the corresponding text and legend have been updated accordingly.

Line 259: “...we observed a clustering of genus belonging to Firmicutes and Actinobacteria, revealing potential similar media preferences.” I noted in my previous review that no such clustering was apparent in Figure 5B and the authors responded “We believe that the statement regarding the proximity of Firmicutes and Actinobacteria in the clustering is supported...” While species of these phyla do both occur in the same very large cluster, they are spread out and intermingled with members of other phyla which I don’t think is well described by “clustering” – maybe a rephrasing like “...we observed that genera belonging to Firmicutes and Actinobacteria were all part of the same large cluster, revealing...” would be more clear.

Response:

We thank the reviewer for giving further clarification and agree that the original phrasing could imply a more distinct clustering of Firmicutes and Actinobacteria than is evident in Figure 5B. To clarify, we have revised the text to in line 283:

“...we observed that genera belonging to Firmicutes and Actinobacteria were all part of the same large cluster, revealing potential similarities in media preferences among taxa from these phyla.”

This should more accurately describe the distribution within the large cluster while preserving the intended point about media preferences.

Figure 5A: The authors responded to my query by confirming the plot is “a standard boxplot with individual data points displayed on the right side,” but did not clarify this in the manuscript itself.

Response:

We thank the reviewer for highlighting this omission. We have now updated the figure legend for Figure 5A to explicitly state that it is a standard boxplot, with individual data points displayed on the right side of the plot to provide additional context for the variation in growth profiles. This should provide clarity in the manuscript.

Lines 499-512: I continue to find the description of the ranking and selection process inadequate. Maybe this descriptive text could be supplemented by a formula indicating the actual calculation? And perhaps one of the examples, for example the Colinsella story, could be fully explained in the supplementary material with real numbers?

Response:

To further clarify to the reader how the combinations were designed, we have updated the method section to include formulas. Additionally, we also created a github repository to share code relevant to the manuscript, including an example of the code useful for designing combinations enriching for Colinsella.

“Briefly, the design of targeted media was done in 4 steps (Supplementary Figure 2). First, all possible combinations of media were generated for the conditions used in the single modification cultivations, up to 6 conditions per combination, amounting to 10.9M combinations. Then, metrics were calculated to score what could be the most likely combinations to yield an enrichment for the target:

1. *Mean relative abundance per OTU/Function (MRA_{target})* The mean relative abundance of the target taxon or function across the modifications included in the combination was calculated using the formula $MRA_{target} = \frac{1}{n} \sum_{i=1}^n RA_{target,i}$, where n is the number of

modifications in the combination and $RA_{target,i}$ is the relative abundance of the target in modification i .

2. Total Mean Relative Abundance of All OTUs/Functions in the Combination (MRA_{total}): The total mean relative abundance of all operational taxonomic units (OTUs) or functions detected across the modifications in the combination was calculated as $MRA_{total} = \frac{1}{n} \sum_{j=1}^m MRA_{OTU_j}$ where m is the total number of OTUs/functions detected across all modifications in the combination and MRA_{OTU_j} is the mean relative abundance of OTU/function j .

3. Ratio of Target Mean Relative Abundance to Total Mean Relative Abundance ($R_{target/total}$): This ratio indicates the proportion of the target's mean relative abundance relative to the total mean relative abundance of all OTUs/functions in the combination and is represented as $R_{target/total} = \frac{MRA_{target}}{MRA_{total}}$

Finally, combinations were ranked for each target based on highest $R_{target/total}$ > lowest total number of OTUs/function m > highest mean relative abundance of the target MRA_{target} . The top 20 modifications for all the combinations for a target were retained. Multiple visualizations were generated in a dashboard to assist in choosing the combinations that could be valuable to test: co-occurrence network of the most present modifications in the best combinations across donors, heatmap of the best modifications relative abundance per donor and the mean relative abundance of the target relative to the number of OTUs/functions in the combination. An example of the code for this process is shared on github at <https://github.com/Jerarm/Metagenome-guided-culturomics-for-the-targeted-enrichment-of-gut-microbes>.”

Lines 291-298 and Figure 6D: In my previous review I asked whether abundance of dopamine and dopac pathways was assessed directly from metagenome data or inferred based on species abundance, and the methods were clarified to make it clear that this analysis was based on functional gene abundance in the metagenomes. Yet the legend for Figure 6D still says “Bar plot for the fold change enrichment over Base for strains associated with dopamine and DOPAC synthetic pathways...” This should say “genes,” not “strains.”

Response:

We thank the reviewer for pointing out this discrepancy. We agree that the correct term should be “genes” rather than “strains,” as the analysis was based on functional gene

abundance in the metagenomes. We have updated the legend for Figure 6D to reflect this, and it now reads:

“Bar plot for the fold change enrichment over Base for genes associated with dopamine and DOPAC synthetic pathways.”

One other minor comment - Line 125: “...were obtained for both cultured communities, as well as the original stool sample.” “Both” is redundant here.

Response:

We agree with the reviewer and have removed the redundant use of “both” in the sentence

We sincerely thank the reviewers for their detailed and insightful comments, which have contributed to improving the quality of our manuscript. In response to their feedback, we have made final revisions to the manuscript. A detailed, point-by-point response is provided below, with our responses in blue.

REVIEWERS' COMMENTS

Reviewer #2 (Remarks to the Author):

My concerns have largely been addressed. Thank you!

There remains one point of disagreement concerning Figure 3. In my original review, I asked why values for the categories "Total on media and stool" and "Total on media only" were present in Figure 3A (OTU richness) but absent from Figure 3B (Faith's PD). I asked this because the plotted entities are the same across Figures 3A and 3B (each point is a Donor) and because the measures – OTU richness and Faith's PD – can each be calculated on any sub-set of taxa.

The authors initially replied that for "Total on media and stool" and "Total on media only", there was "no value" for Figure 3B because Faith's PD would "not be meaningful to calculate" for these two categories. I did not fully understand this reply, because the only requirement for calculating Faith's PD on a sub-set of taxa is that all taxa appear together in a phylogenetic tree. The authors display this tree in Figure 2. Thus, upon re-review, I asked for further clarification.

The authors have now replied that Faith's PD must be calculated on a "single community or sample", that calculating Faith's PD is "not appropriate" in these cases because taxa are aggregated across samples, and that Faith's PD calculation "assumes that taxa are part of a single, cohesive ecological community where phylogenetic relationships are meaningful in the context of the community", and that doing so would "violate this assumption, making such PD calculations statistically invalid and their interpretation unreliable". And so on – e.g., Faith's PD "require[es] taxa to be part of a coherent ecological unit for phylogenetic relationships to be meaningful".

These statements are false. Phylogenetic relationships are not context- (or community-) dependent and Faith's PD makes no such assumption. In fact, the paper in which Daniel Faith introduces the metric (Faith 1992) is concerned with conservation priorities, reserve design, and assessing the phylogenetic diversity of sub-sets of threatened species – not community ecology. I refer the authors to this paper below. Faith's PD is the sum of branch length leading to a set of taxa in a phylogenetic tree. It is not an estimator; not an index. What is true – OTU richness and Faith's PD may not agree; they may not yield the same pattern or conclusion. Two sets of taxa may have identical OTU richness (say, 5 taxa each) but very different Faith's PD because one set has five very closely related taxa while the other has five very distantly related ones. Thus, comparing the two metrics may yield interesting insights into the nature of diversity.

If the authors do not think it reasonable or meaningful to assess alpha diversity for sets of taxa representing more than one sample or community, that's fine. (But are they even doing that? The community exists at the level of the Donor.) It just doesn't make sense to show one metric and not the other. It would seem reasonable to me to show both metrics or neither metric for the categories "Total on media and stool" and "Total on media only".

Faith, Daniel P. "Conservation Evaluation and Phylogenetic Diversity." *Biological Conservation* 61, no. 1 (1992): 1–10. [https://doi.org/10.1016/0006-3207\(92\)91201-3](https://doi.org/10.1016/0006-3207(92)91201-3).

R= We appreciate the reviewer's detailed feedback and the opportunity to clarify our methodology and data presentation in Figure 3. In Figure 3A, we present species richness for each donor across different media modifications. We also include cumulative counts for 'total on media only' and 'total on media & stool' samples. Calculating total species richness for combined samples is meaningful because we aim at 1. providing insights into the overall taxonomic diversity captured from each donor throughout the study and 2. help evaluate how well our cultivation methods recovered the donors' microbiota diversity compared to the original stool samples. Species richness is a simple count of unique taxa and does not rely on phylogenetic relationships or assume ecological interactions among taxa.

However, to ensure clarity and avoid any potential confusion for readers, we have removed the categories "Total on media and stool" and "Total on media only" from the figure and reference in the main text. This adjustment simplifies the presentation and keeps the focus on the key aspects of the study, ensuring the manuscript remains clear and impactful for readers.

Reviewer #3 (Remarks to the Author):

I believe my remaining concerns have largely been addressed. I am still concerned that the observations in lines 178-188 are statistically unsupported; while the wording has been toned down I think it would still benefit the reader to include p-values or note that additional experiments would be needed to validate the trends observed. For example, the paragraph could start, "In the aromatic amino acids group, some qualitative observations that merit further exploration are that histidine and tryptophan..." I have not gone through the manuscript sentence by sentence to make sure all assertions are statistically supported, but I encourage the authors to make sure this is the case.

R= We have revised the manuscript and taken care to tone down any claims that are not statistically supported. Wherever possible, we have clarified that additional experiments would be needed to validate the observed trends. In particular, as suggested by the reviewer we have rephrased the section in lines 160-164:

"In the aromatic amino acids group, we observed that histidine and tryptophan tended to enrich for more diverse taxa than tyrosine, with tryptophan appearing to selectively favor Clostridiaceae, Bifidobacteriaceae, and Akkermansiaceae species compared to the base medium (as suggested by PD values in Figure 3B and the number of green squares in Figure 3C). However, these observations are only qualitative and merit further exploration."